# Semantic-Guided Representation Enhancement for Self-supervised Monocular Trained Depth Estimation

## Abstract

Self-supervised depth estimation has shown its great effectiveness in producing high quality depth maps given only image sequences as input. However, its performance usually drops when estimating on border areas or objects with thin structures due to the limited depth representation ability. In this paper, we address this problem by proposing a semantic-guided depth representation enhancement method, which promotes both local and global depth feature representations by leveraging rich contextual information. In stead of a single depth network as used in conventional paradigms, we propose an extra semantic segmentation branch to offer extra contextual features for depth estimation. Based on this framework, we enhance the local feature representation by sampling and feeding the point-based features that locate on the semantic edges to an individual *Semantic-guided Edge Enhancement module (SEEM)*, which is specifically designed for promoting depth estimation on the challenging semantic borders. Then, we improve the global feature representation by proposing a semantic-guided multi-level attention mechanism, which enhances the semantic and depth features by exploring pixel-wise correlations in the multi-level depth decoding scheme. Extensive experiments validate the distinct superiority of our method in capturing highly accurate depth on the challenging image areas such as semantic category borders and thin objects. Both quantitative and qualitative experiments on KITTI show that our method outperforms the state-of-the-art methods.

## 1 Introduction

Depth estimation is a long standing problem in computer vision community, which offers useful information to a wide range of tasks including robotic perception, augmented reality and autonomous driving, *etc.* Compared with depth estimation methods which rely on active vision or multi-view paradigms (Schonberger & Frahm, 2016; Li et al., 2019), estimating depth from only a single image is highly ill-posed, thus brings greater challenges for higher quality results.

In recent years, monocular depth estimation has witnessed new renaissances with the advent of deep learning (He et al., 2016; Jaderberg et al., 2015; Simonyan & Zisserman, 2014). By learning deep representations in a supervised manner, various of networks (Eigen & Fergus, 2015; Eigen et al., 2014; Laina et al., 2016) are capable of producing high quality depth maps thanks to the large corpus of training data. At the mean time, consider the lack of labeled training data for network training, recent advances (Zhou et al., 2017; Godard et al., 2019; 2017) show that monocular depth estimation can be accomplished in a self-supervised way. The network can be trained by unlabeled image sequences using two-view geometric constraints, while achieving comparable results with the supervised paradigm. The learning-based methods managed to handle the highly ill-posed monocular depth estimation problem by implicitly learning the mapping between visual appearance and its corresponding depth value.

However, despite the great effectiveness for learning-based depth estimation, these methods still struggle to conduct precise depth estimation on challenging image regions such as semantic category borders or thin object areas. For example, the estimated object depth usually fails to align with the real object borders, and the depth of foreground objects which have thin structures tends to be

submerged in the background. We attribute these phenomena to the limited depth representation ability that (1) the pixel-wise local depth information can not be well represented by current depth network, especially on highly ambiguous, semantic border areas, (2) current depth representations are not capable of well describing depth foreground/background relationships globally. These issues lead to the wrong depth estimation from the true scene structures, which hinders the further applications in the real-world tasks.

In this paper, we address this problem via enhancing both local and global depth feature representations for self-supervised monocular depth estimation via semantic guidance. As semantic segmentation conducts explicit category-level scene understandings and produces well-aligned object boundary detection, we propose an extra semantic estimation branch inside the self-supervised paradigm. The semantic branch offers rich contextual features which is fused with depth features during multiscale learning. Under this framework, we propose to enhance the depth feature using semantic guidance in a *local-to-global* way. To improve the local depth representations on semantic category borders, inspired by the sampling and enhancing strategies used in semantic segmentation (Kirillov et al., 2020), **our first contribution** is to propose a *Semantic-guided Edge Enhancement Module* (SEEM) which is specially designed to enhance the local point-based depth representations that locate on the semantic edges. Different from the method in (Kirillov et al., 2020), we enhance the point features using multi-level of representations from different domains under a self-supervised framework. To be specific, we sample a set of point positions lying on the binary semantic category borders, and extract the point-wise features of the corresponding edge positions from the encoded feature, depth decoding feature as well as the semantic decoding feature. We then merge and enhance the point-wise features and feed them to the final depth decoding features to promote the edge-aware depth inference for self-supervision. For global depth representation enhancement, **our second contribution** is to propose a semantic-guided multi-level attention module to improve the global depth representation. Different from the conventional self-attentions (Fu et al., 2019; Vaswani et al., 2017) which are implemented as single modules on the bottleneck feature block, we propose to explore the self-attentions on different level of decoding layers. In this way, both semantic and depth representations can be further promoted by exploring and leveraging the pixel-wise correlations inside of their fused features.

We validate our method mainly on KITTI 2015 (Geiger et al., 2012), and the Cityscapes (Cordts et al., 2016) benchmark is also used for evaluating the generalization ability. Experiments show that the proposed method significantly improves the depth estimation on category edges and thin scene structures. Extensive quantitative and qualitative results validate the superiority of our method that it outperforms the state-of-the-art methods for self-supervised monocular depth estimation.

## 2 RELATED WORK

There exist extensive researches on monocular depth estimation, including geometry-based methods (Schonberger & Frahm, 2016; Enqvist et al., 2011) and learning-based methods (Eigen et al., 2014; Laina et al., 2016). In this paper, however, we concentrate only on the self-supervised depth training and semantic-guided depth estimation, which is highly related to the research focus of this paper.

**Self-supervised depth estimation.** Self-supervised methods enable the networks to learn depth representation from merely unlabeled image sequences that they reformulate the depth supervision loss into the image reconstruction loss. Godard et al. (2017) and Garg et al. (2016) first propose the self-supervised method on stereo images, then Zhou et al. (2017) propose a monocular trained approach using a separate motion estimation network. Based on these frameworks, a large corpus of works seek to promote self-supervised learning from different aspects. For more robust self-supervision signals, Mahjourian et al. (2018) propose to use 3D information as extra supervisory signal, another kind of methods leverage additional information such as the optical flow (Ranjan et al., 2019; Wang et al., 2019b) to strengthen depth supervision via consistency constraints. In order to solve the loss deviation problems on non-rigid motion areas, selective masks are used to filter out the these areas when computing losses. Prior works generate the mask by the network itself (Zhou et al., 2017; Yang et al., 2017; Vijayanarasimhan et al., 2017), while the succeeding methods produce the mask by leveraging geometric clues (Bian et al., 2019; Wang et al., 2019a; Godard et al., 2019; Klingner et al., 2020), which is proved to be more effective. There also exist other methods trying to enhance the network performance with traditional SfM (Schonberger & Frahm, 2016), which offer pseudo labels for depth estimation. Guizilini et al. (2020a) propose a novel network architecture to improve depth estimation. In this paper, we do not consider the

performance improvements contributed by this novel architecture in experimental comparisons, and compare different methods with conventional backbones.

**Depth estimation using semantics.** Semantic information are shown to provide positive effects toward depth estimation framework in the previous works. The methods can be categorized into two groups by their way to use the semantic information. One group of the methods leverage the semantic labels directly to guide the depth learning. The works of Ramirez et al. (2018) and Chen et al. (2019a) propose depth constraints by leveraging the semantic labels of the scene. Klingner et al. (2020) and Casser et al. (2019) address the non-rigid motion issues by handling the moving object areas highlighted by the semantic map. Wang et al. (2020) use a divide-and-conquer strategy to conduct depth estimation with different semantic categories, and Zhu et al. (2020) offer a depth morphing paradigm with the help of semantic foreground edges. The other group of methods enhance the depth representation by feature manipulation. Chen et al. (2019a) generated depth and semantic maps by a single latent representation. Ochs et al. (2019) proposed a segmentation-like loss item for depth estimation. Guizilini et al. (2020b) leveraged a pre-trained semantic segmentation network and further conduct feature merging by PAC-based (Su et al., 2019) module.

In this paper, we propose an individual semantic network for depth estimation. But different with Guizilini et al. (2020b), our semantic branch shares the same encoder and is trained together with the depth network for better scene representation. We leverage the semantic information in both ways. The semantic edges are used to guide the edge-based point feature sampling for local depth depth representation enhancement. At the mean time, we utilize the semantic feature blocks to conduct multi-scale feature fusion for global depth representation enhancement. The proposed method constructs a holistic way for feature representation enhancing in self-supervised depth estimation.

## 3 SELF-SUPERVISED DEPTH ESTIMATION FRAMEWORK

The proposed method builds upon the self-supervised depth estimation framework. An image triplet $(I_{t-1}, I_t, I_{t+1})$ is used as the input, where $I_t$ acts as the target image and $I_{t'} \in \{I_{t-1}, I_{t+1}\}$ are the source images. During training, the target image $I_t$ is fed into the depth network $f_D$ to get the estimated depth $D_t = f_D(I_t)$, while the adjacent image pairs $(I_t, I_{t'})$ are put into the ego-motion network for the 6-DoF ego-motion estimations $T_{t' \to t}$. Then, the synthesized target images $I_{t' \to t}$ can be estimated using the source images $I_{t'}$, depth $D_t$ and the ego-motions $T_{t' \to t}$, following the formulation from Zhou et al. (2017). The self-supervised loss can be calculated by

$$L(I_t, I_{t' \to t}) = L_p(I_t, I_{t' \to t}) + \lambda L_s(I_t, I_{t' \to t}), \tag{1}$$

where $\lambda$ is the weighting factor between the photometric loss $L_p$ and smoothness loss $L_s$. Following Godard et al. (2019), we implement the minimum photometric loss

$$L_p(I_t, I_{t' \to t}) = \min_{t'} (\frac{\alpha}{2} \left(1 - \text{SSIM}\left(I_t, I_{t' \to t}\right)\right) + (1 - \alpha) \left\|I_t - I_{t' \to t}\right\|_1), \tag{2}$$

where SSIM is the Structural Similarity item (Wang et al., 2004), $\alpha$ is the weighting factor which is set to $0.85$. The smoothness loss item (Godard et al., 2017) is implemented to smooth the depth map by the consistency between the image and depth gradient

$$L_s(I_t) = |\partial_x D_t| \, e^{-|\partial_x I_t|} + |\partial_y D_t| \, e^{-|\partial_y I_t|}, \tag{3}$$

where $\partial_x, \partial_x$ denote gradient operation on x,y-axis. In our implementation, we also conduct auto-masking and multi-scale depth upsampling strategy as proposed in Godard et al. (2019) to further improve depth estimation.

## 4 THE PROPOSED METHOD

In this paper, we promote the self-supervised depth estimation paradigm via proposing a semantic-guided representation enhancing depth network. The overview of the method is shown in Figure 1. We propose a multi-task framework which consists of the depth and semantic decoding branch with shared feature encoder. During the training stage, the semantic branch feeds contextual features to the depth branch to enhance depth comprehension. Under this multi-task framework, we propose a Semantic-guided Edge Enhancement Module (SEEM) to enhance the local depth feature representation on semantic category edges. Meanwhile, we enhance the global depth feature representations by

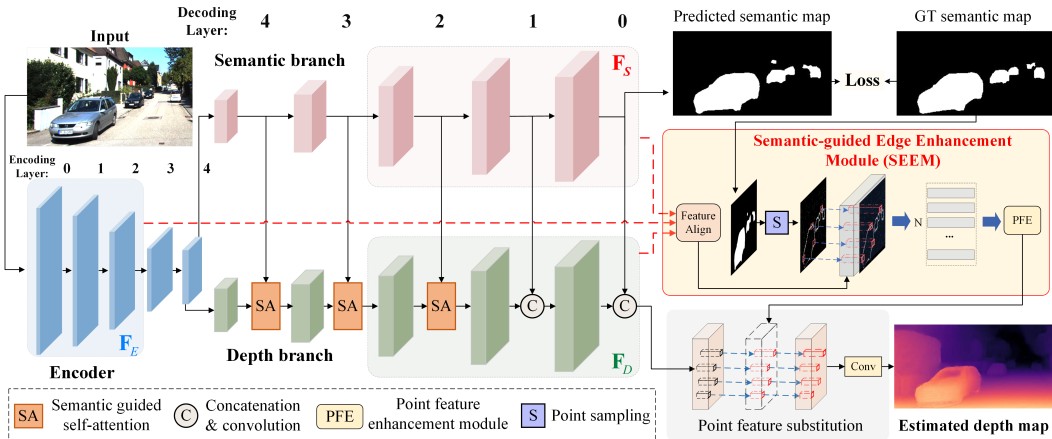

Figure 1: **The overview of the proposed architecture**. An extra semantic branch is proposed to offer contextual information for depth representations. Then, we enhance the local depth feature representation via sampling and enhancing semantic edge-based point features via the proposed *Semantic-guided Edge Enhancement module* (SEEM). The global feature representation is promoted by the proposed semantic-guided multi-layer self-attention (SA), which explores pixels feature correlations between fused depth and semantic features.

proposing the semantic-guided multi-level self-attention module, which improves both the semantic and depth feature representations by digging into the pixel-wise dependencies.

## 4.1 THE PROPOSED SEMANTIC SEGMENTATION BRANCH

We extend the general depth network with an extra semantic segmentation branch to provide explicit contextual information for depth estimation. The semantic branch shares the same network structure and input feature blocks with the depth branch. During network training, it feeds the semantic features to the depth network in a multi-layer way, to facilitate better depth representation and used for further depth representation enhancement processing. Different from general segmentation tasks, given input image $I_t$, it outputs the binary semantic probability map $\hat{M}_t^b$, which implicitly encodes distance information by assigning foreground and background areas. This setting is empirically proved to offer abundant guidances for better depth estimation, while leading to less computational burden. Details can be found in Section C.4 of the Appendix. Consider we build our method on the self-supervised scenarios, we use the semantic pseudo label generated by off-the-shelf method (Zhu et al., 2019) to supervise the semantic branch. We use binary cross-entropy loss to supervise the semantic branch. The semantic loss item is $L_m(\hat{M}_t^b, M_t^b)$, where $M_t^b$ is the binary semantic pseudo label.

## 4.2 SEMANTIC-GUIDED EDGE ENHANCEMENT MODULE (SEEM)

We improve depth estimation on semantic category edges via sampling and enhancing the local point features through the proposed *Semantic-guided Edge Enhancement Module* (SEEM). As shown in Figure 2, SEEM takes the encoding feature blocks $F_E$, depth decoding feature blocks $F_D$ as well as the semantic decoding feature blocks $F_S$ as input, and output a set of point features $F' \in \mathbb{R}^{N \times C}$, where $N$ is the number of sampled points, $C$ denotes the output feature channel which is consistent with the final output depth features. We first sample the local points according to the semantic edges. Then, we extract the point features from input feature blocks via grid-sampling, and enhance these feature representations via merging and feeding them into a point feature enhancement module. The resulting representative point features are used to replace the final output depth features on the corresponding positions, to further promote the depth representations on semantic category edges.

**Semantic-guided edge point sampling strategy.** A vanilla sampling strategy is to sample all points directly on semantic edges. However, it shows less robustness that (1) the semantic edge is produced by pseudo labels instead of GT labels, so it may not sample the real edges, and (2) the points nearby the semantic edges are also challenging areas for depth estimation, which should also be taken into consideration. To address these issues, we propose a sampling strategy which incorporates enough redundancies. We first compute the semantic edges response via convolving the binary semantic

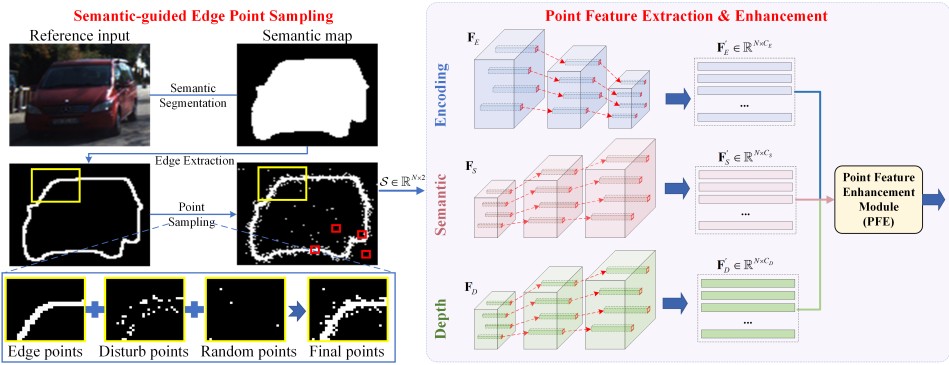

Figure 2: **The semantic-guided edge enhancement module (SEEM)**. The proposed method first samples and extracts the semantic edge-located point set which consists of the edge points, disturbed edge points and random points. Then, the points features are extracted via bilinear sampling and fed into the point feature enhancement module which includes a set of 1-D convolutions, to enhance the local point depth feature representations independently.

mask with the Sobel operator, and then extract the edges points set $\mathcal{S}_{\mathcal{E}} \in \mathbb{R}^{\mu N \times 2}$ which consists of $\mu N$ points that exhibit largest responses from the edge response map. Besides the points lying exactly on the edges, we choose another set of $\mu N$ uniformly sampled random points in a small range $[-c, c]$, and use this random point set to disturb the point positions of $\mathcal{S}_{\mathcal{E}}$ to get a new disturbed edge point set $\mathcal{S}_{\mathcal{D}} \in \mathbb{R}^{\mu N \times 2}$, in order to sample the real ground truth edge points and nearby points. Finally, a point set $\mathcal{S}_{\mathcal{R}} \in \mathbb{R}^{(1-\mu N) \times 2}$ with $(1 - 2\mu)N$ points randomly sampled on the global image is selected to generalize the network's learning abilities. The final point set $\mathcal{S} \in \mathbb{R}^{N \times 2}$ is shown in Equation 4, where $\cup$ is the union operator.

$$\mathcal{S} = \mathcal{S}_{\mathcal{E}} \cup \mathcal{S}_{\mathcal{D}} \cup \mathcal{S}_{\mathcal{R}}, \tag{4}$$

**Point feature extraction and enhancement.** For the input feature block among $F_E, F_D, F_S$, they are free to contain flexible layers of features. In this paper, we select features of layer $(2, 1, 0)$ (marked with darker boxes in Figure 1) that they are regarded to offer abundant information for image, depth, and semantic representations. We extract the point features via the point set $\mathcal{S}$

$$F'_j = F_j(\mathcal{S}), j \in \{E, D, S\}. \tag{5}$$

In Equation 5, $F'_j \in \mathbb{R}^{N \times C_j}, j \in \{E, D, S\}$ denotes the extracted point features from the three input features blocks. In each input feature block, point feature extraction across different size of feature maps is conducted by bilinear interpolation (Jaderberg et al., 2015). We then feed these point-wise feature representations from different domains (*e.g.*, image, geometry and semantic domains) to a point feature enhancement (PFE) module, which first concatenates the input features in channel-wise manner, and then enhances point representations via a set of 1-D convolutions with kernel size of 1. The output is the point feature block $F'_{out} \in \mathbb{R}^{N \times C}$

$$F'_{out} = f_\theta(F'_E, F'_D, F'_S), \tag{6}$$

where $f_\theta$ denotes the feature enhancement network, the output dimension $C$ is consistent with the last output features of the depth branch. We then substitute the depth feature with enhanced point feature representations $F'_{out}$ on the corresponding position of $\mathcal{S}$. Since the module is specially designed for enhancing point-wise depth feature representations on semantic border areas, with the help of multi-domain feature representations, it's capable of conducting better depth estimation as is shown in the experiment section.

### 4.3 SEMANTIC-GUIDED MULTI-LEVEL SELF-ATTENTION

In the scene perception task, the depth information is indeed, implicitly reflecting the contextual information in the scene. For instance, some semantic category areas such as cars or pedestrians often act as the foreground, thus they consistently exhibit smaller depth than their surrounding pixels. In our method, the semantic segmentation branch provides contextual information on each decoding layer of the depth branch. Consider the inherent correlation between depth and semantic representations as analysed above, we propose the multi-level self-attention which explores the pixel-wise

correlation inside the fused depth and semantic feature, to further improve the global feature representations for better depth estimation.

Conventional self-attention mechanisms are usually implemented on the bottleneck features as a single module. In this paper, we propose the semantic-guided multi-level attention which instead consists of a set of multi-level self-attentions that locate on the multi-level decoding layers. The attention is conducted on both depth and semantic features. In the depth decoding stage, suppose $F_D^i$ and $F_S^i$ the i-th level of feature maps from the depth and semantic branch, we concatenate the two features and fuse them via a set of convolutions to get the feature map $F_A^i$. For a given fused feature $F_A^i \in \mathbb{R}^{C_i \times H_i \times W_i}$, the self-attention module calculates the query feature, key feature and value feature $\{Q_i, Y_i, V_i\} \in \mathbb{R}^{C_i \times L_i}$ via convolution operation. $H_i$ and $W_i$ are height and width of features on the i-th level, and $L_i = H_i \times W_i$. The point-wise attention on the i-th level can be calculated by

$$A_i = softmax(Q_i \cdot Y_i^T). \tag{7}$$

Finally, the output feature can be formulated as the weighted matrix product of the value feature and the attention, added by the residual input feature

$$F_{out}^i = \gamma(A_i \cdot V_i) + F_A^i, \tag{8}$$

where $\gamma$ is a learnable weight factor which is initially set to 0. In our paper, we implement multi-level attentions on 3 decoding layers (layer 4, 3, 2) which has smaller feature resolutions. Detailed illustrations can be found in the Appendix.

### 4.4 FINAL LOSS

Based on the proposed method, the final loss can be formulated as

$$L_{final} = L_p + L_m + \omega \cdot L_s, \tag{9}$$

where $\omega$ is set to 0.001 following common practice (Godard et al., 2019; 2017; Zhou et al., 2017), $L_m$ denotes the semantic loss as illustrated in Section 4.1.

## 5 EXPERIMENTS

In this section, we conduct extensive experiments to validate the superiority of the proposed method. We use standard KITTI benchmark (Geiger et al., 2012) to train and evaluate the proposed network. The Eigen's split (Eigen et al., 2014) along with Zhou's pre-processing scheme (Zhou et al., 2017) is selected which leads to 39810 images for training, 4424 images for validation and 697 images for testing. We also use the Cityscapes dataset (Cordts et al., 2016) to evaluate the generalization ability of the proposed method. The dataset is directly tested on its test split which consists of 1525 images.

### 5.1 IMPLEMENTATION DETAILS

**Details of model training and testing.** Our method is build upon Monodepth2 (Godard et al., 2019), with ResNet-50 (He et al., 2016) initialized by the ImageNet (Deng et al., 2009) as the backbone. The code is implemented with Pytorch (Paszke et al., 2017) and trained with 1 NVIDIA Tesla V100 GPU. The input image size is $192 \times 640$. The motion network and depth branch are trained with self-supervision. During the point sampling process, we set $N = 3000$ that the number is enough to cover most of the edge areas on the images, $\mu$ is set to $0.4$ by calculating the average number of edge points from randomly selected $10000$ images on KITTI, and $c$ is set to 3 empirically. The network is trained with batch size 12 for 19 epochs, with learning rate of $10^{-4}$ divided by 10 after 15 epochs. The final model is selected by evaluating the models run with random seeds, using the validation set.

**Details of semantic label generation.** For the semantic branch, the network is trained with the pseudo label dataset generated by the off-the-shelf model (Zhu et al., 2019) (mIoU of 80% on Cityscapes validation set). In our paper, the semantic model $M_{CSV+K}$ is trained on Cityscapes (Cordts et al., 2016) and Mapillary Vistas dataset (Neuhold et al., 2017), and fine-tuned with KITTI Semantic (Geiger et al., 2012) (contains 200 training images) using 10-split cross validation strategy. Compared with the large quantity of KITTI raw data used for self-supervision (more than 39,000 training items), the required labeled semantic dataset only accounts for a very tiny proportion, which indicates a relatively lower cost. To further validate the generalization of our method trained with

Table 1: **Quantitative results on KITTI 2015**. The best results are in **bold** and the second best results are underlined. 'S' and 'M' refer to self-supervision methods using stereo images and monocular images, respectively. 'Inst' and 'Sem' mean methods that leverage instance or semantic segmentation information. 'PN' and 'R50' refer to the method that uses PackNet (Guizilini et al., 2020a) and Resnet-50 as backbone, respectively. All monocular trained methods are reported without post-processing steps. The metrics marked in blue mean 'lower is better', while these in red refer to 'higher is better'. Our method outperforms the state-of-the-arts in most metrics by a large margin.

| Method | Training | Abs Rel | Sq Rel | RMSE | RMSE$_{log}$ | $\delta < 1.25$ | $\delta < 1.25^2$ | $\delta < 1.25^2$ |
|---|---|---|---|---|---|---|---|---|
| Godard et al. (2017) | S | 0.133 | 1.142 | 5.533 | 0.230 | 0.830 | 0.936 | 0.970 |
| Pillai et al. (2019) | S | 0.112 | 0.875 | 4.958 | 0.207 | 0.852 | 0.947 | 0.977 |
| Watson et al. (2019) | S | 0.106 | 0.780 | 4.695 | 0.193 | 0.875 | 0.958 | 0.980 |
| Godard et al. (2019) | MS | 0.106 | 0.806 | 4.630 | 0.193 | 0.876 | 0.958 | 0.980 |
| Zhou et al. (2017) | M | 0.183 | 1.595 | 6.709 | 0.270 | 0.734 | 0.902 | 0.959 |
| Mahjourian et al. (2018) | M | 0.163 | 1.240 | 6.220 | 0.250 | 0.762 | 0.916 | 0.968 |
| Yin & Shi (2018) | M | 0.155 | 1.296 | 5.857 | 0.233 | 0.793 | 0.931 | 0.973 |
| Wang et al. (2018) | M | 0.151 | 1.257 | 5.583 | 0.228 | 0.810 | 0.936 | 0.974 |
| Ranjan et al. (2019) | M | 0.140 | 1.070 | 5.326 | 0.217 | 0.826 | 0.941 | 0.975 |
| Chen et al. (2019b) | M | 0.135 | 1.070 | 5.230 | 0.210 | 0.841 | 0.948 | 0.980 |
| Godard et al. (2019) | M | 0.115 | 0.903 | 4.863 | 0.193 | 0.877 | 0.959 | 0.981 |
| Guizilini et al. (2020a) | M | 0.111 | 0.785 | 4.601 | 0.189 | 0.878 | 0.960 | 0.982 |
| Johnston & Carneiro (2020) | M | 0.106 | 0.861 | 4.699 | 0.185 | 0.889 | 0.962 | 0.982 |
| Casser et al. (2019) | M+Inst | 0.141 | 1.026 | 5.291 | 0.215 | 0.816 | 0.945 | 0.979 |
| Chen et al. (2019a) | M+Sem | 0.118 | 0.905 | 5.096 | 0.211 | 0.839 | 0.945 | 0.977 |
| Guizilini et al. (2020b)-(PN) | M+Sem | 0.102 | 0.698 | 4.381 | 0.178 | 0.896 | 0.964 | 0.984 |
| Guizilini et al. (2020b)-(R50) | M+Sem | 0.113 | 0.831 | 4.663 | 0.189 | 0.878 | **0.971** | **0.983** |
| **Ours** | M+Sem | **0.105** | **0.764** | **4.550** | **0.181** | **0.891** | 0.965 | **0.983** |

semantic labels without KITTI groundtruth guidance, we offer another semantic dataset generated by $M_{CSV}$, which is trained on Cityscapes and Mapillary Vistas dataset, without involving any images from KITTI. The results are shown in Section 5.4. During semantic dataset generation, the raw KITTI data is fed into the pre-trained semantic model (Zhu et al., 2019) to get the fully segmented images. Then, we get the binary semantic labels by assigning the traffic signs, different type of vehicles, and people as foreground. Details can be found in Table 7 of the Appendix.

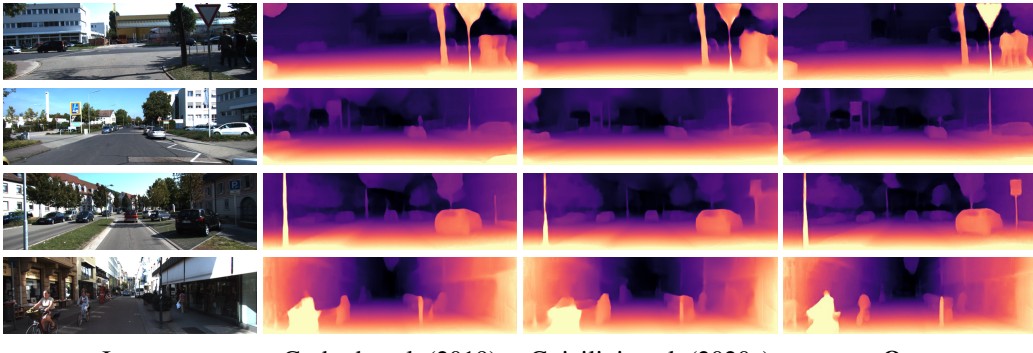

| Input | Godard et al. (2019) | Guizilini et al. (2020a) | **Ours** |

Figure 3: **Qualitative comparison on KITTI 2015**. Our method shows its distinct advantages against other methods in generating sharp depth borders as well as thin structures that is well-aligned with the semantic contextual information.

## 5.2 KITTI RESULTS

We compare our method with state-of-the-art self-supervised methods including stereo-trained methods, monocular-trained methods as well as other semantic-guided methods. The image resolution of the latest methods is set to $192 \times 640$. For fair comparison, we evaluate the latest semantic-guided method (Guizilini et al., 2020b) with its ResNet-50 backbone setting instead of its PackNet setting, to rule out the performance improvement benefited from better network architecture. Quantitative results are shown in Table 1. We observe that the proposed method outperforms all the current state-of-the-art self-supervised methods by a substantial margin, even including the stereo trained methods under the same input settings. The qualitative results are shown in Figure 3. We can see that compared with other recent state-of-the-art methods (Godard et al., 2019; Guizilini et al., 2020a), our method shows its remarkable advantages that our method (1) precisely recovers

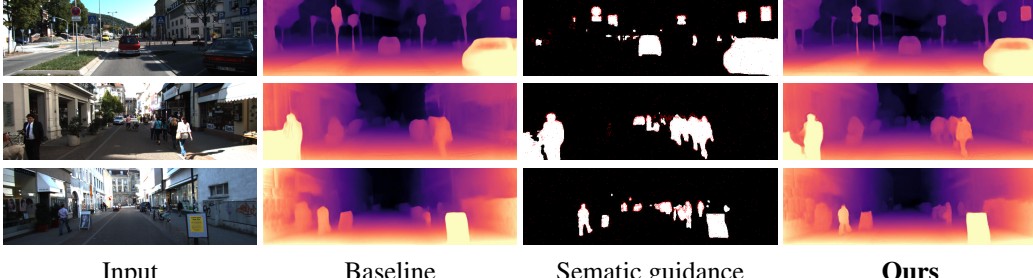

| Input | Baseline | Sematic guidance | **Ours** |

Figure 4: **Qualitative results of the proposed method against the baseline**. The third column refers to the semantic guidance, in which the black and white area denote the foreground and background, and the red dots are the sampled points of the proposed SEEM module.

the depth contours of the semantic objects such as humans or traffic signs, even thin structures such as human legs or traffic sign poles can also be well estimated, (2) Our method precisely distinguishes between the depth foreground and background, which lead to better estimations of the global scene structures. More results can be found in the Appendix.

Table 2: **Ablation experiments**. Results of several versions of our proposed method on KITTI 2015 (Geiger et al., 2012) are reported. The best results are in **bold** and the second best are underlined.

| Method | SEEM | SA | Abs Rel | Sq Rel | RMSE | $RMSE_{log}$ | $\delta \leq 1.25$ | $\delta \leq 1.25^2$ | $\delta \leq 1.25^3$ |
|---|---|---|---|---|---|---|---|---|---|
| Baseline | ✗ | ✗ | 0.110 | 0.830 | 4.639 | 0.187 | 0.884 | 0.962 | 0.982 |
| Baseline+SEEM | ✓ | ✗ | 0.107 | 0.771 | 4.579 | 0.184 | 0.887 | 0.963 | **0.983** |
| Baseline+SA | ✗ | ✓ | 0.106 | 0.794 | 4.578 | 0.183 | 0.890 | 0.964 | **0.983** |
| Baseline+SEEM+SA | ✓ | ✓ | **0.105** | **0.764** | **4.550** | **0.181** | **0.891** | **0.965** | **0.983** |

## 5.3 ABLATION STUDIES

To further validate the effectiveness of the proposed contributions, We conduct ablation studies to analyze the performance improvements from individual contributions and their collaboration toward the baseline. The quantitative results are shown in Table 2. We observe that both SEEM module and the semantic-guided multi-level attention show distinct improvements, and their combination consistently improve the performance toward the baseline. We also show the visual improvements of our method in Figure 4. The baseline and the final results are shown in the second and final column. Pseudo semantic guidances are shown in the third column, where white/black denote the foreground/background area, the red dots represents the sampled point positions. We can see from the figure that with the help of semantic guidances, the details of the final depth maps are significantly promoted, especially on the binary semantic edges, as well as the point-sampled areas. It indicates that our proposed contributions successfully leverage the semantic information to recover the detailed depth information on foreground semantic categories, as well as constraining the depth boundaries to be aligned with the semantic object borders.

## 5.4 FURTHER ANALYSIS ON SEMANTICS

**Analysis on different semantic models**. To evaluate the influence brought by semantic models trained in different ways, we train our model with pseudo labels generated by $M_{CSV}$, which does not involve KITTI fine-tuning. Results are shown in Table 3. The segmentation performances of two semantic models $M_{CSV+K}$ and $M_{CSV}$ are evaluated on KITTI Semantics (Geiger et al., 2012) (200 images), both full semantic mIoU (including 19 categories from Cityscapes labels) and binary mIoU (2 categories) are reported. We find an interesting phenomenon that although $M_{CSV+K}$'s performance (79.90%) is obviously better than $M_{CSV}$'s (67.73%) in terms of $mIoU_{Full}$, both methods' performances of $mIoU_{Bi}$ are very close (94.74% VS 93.54%). We attribute it to the reason that false segmentations often occur between categories solely belong to the background (or the foreground). Thus, the binary mask effectively alleviate the performance declines from full semantic labels. This indicates a general advantage which still hold for other models trained with other datasets. Visual details can be found on Figure 8 of the Appendix. Quantitative results of Table 3 show that our model trained on $M_{CSV}$ semantic labels generates comparable results toward that trained on $M_{CSV+K}$ labels. This validates the advantage of using binary semantic map, that the network performance will not be greatly influenced by different pre-trained semantic models.

**Semantic category-level performance**. To further validate the advantage of the semantic guidance, we show the performance improvements on each semantic category toward the baseline method

Table 3: **Performance of models trained on different semantic datasets**. The binary segmentation IoU (mIoU$_{Bi}$) differs a little across different datasets, and our model achieves comparable results when trained on cross-domain generated semantic labels.

| Semantic model | mIoU$_{Full}$(%) | mIoU$_{Bi}$(%) | Abs Rel | Sq Rel | RMSE | RMSE$_{log}$ | $\delta \leq 1.25$ | $\delta \leq 1.25^2$ | $\delta \leq 1.25^3$ |
|---|---|---|---|---|---|---|---|---|---|
| $M_{CSV+K}$ | 79.90 | 94.74 | 0.105 | 0.764 | 4.550 | 0.181 | 0.891 | 0.965 | 0.983 |
| $M_{CSV}$ | 67.73 | 93.54 | 0.105 | 0.768 | 4.571 | 0.182 | 0.890 | 0.964 | 0.983 |

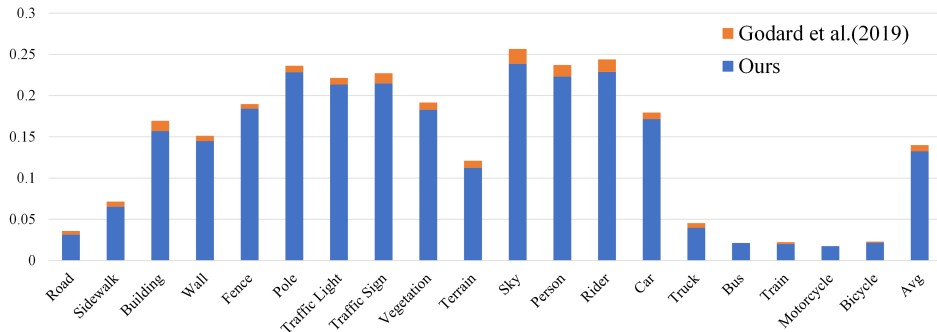

Figure 5: **Semantic category-level improvement**. We use the AbsRel metric for evaluation. Blue refers to our methods and red denotes the baseline method (Godard et al., 2019). Benefited from the proposed SEEM module and semantic-guided multi-level attentions, our method improvements consistently on most of semantic categories.

(Godard et al., 2019). The results are shown in Figure 5. Consider there is no groundtruth semantic labels for KITTI raw, we use the pseudo labels to find semantic categories. As shown in the figure, except for category "Motorcycle" (performances are close with a very small margin of 0.00016), our method improves consistently across all other categories, including foreground category "traffic sign", "person", "car" and background category "sky", "building" and "vegetation". It validates that our method not only learns good representations of border areas, but also benefits from the global semantic contextual information provided by semantic guidance.

## 5.5 CITYSCAPES RESULTS

We further validate the generalization ability of our proposed method on Cityscapes (Cordts et al., 2016) with models trained only on KITTI (Geiger et al., 2012). Compared with the state-of-the-art methods, our method conducts accurate and consistent depth estimation inside object categories, and recover detailed depth information on thin structures from even distant objects. Moreover, during testing, our method consistently conducts high quality predictions using either groundtruth or pseudo semantic guidance as input. Additional results can be found in Figure 10 of the Appendix.

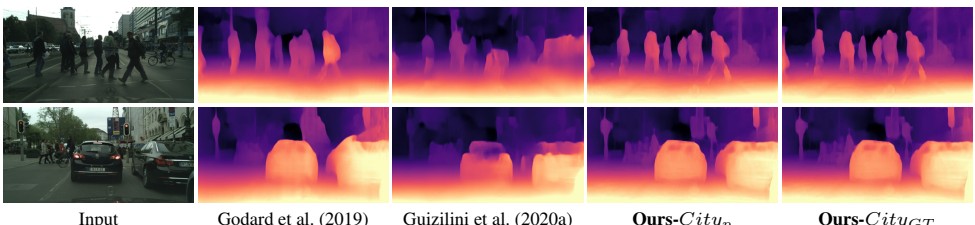

| Input | Godard et al. (2019) | Guizilini et al. (2020a) | **Ours-**$City_p$ | **Ours-**$City_{GT}$ |

Figure 6: **Qualitative results on Cityscapes** (Cordts et al., 2016). All methods are trained on KITTI (Geiger et al., 2012) to evaluate the generalization abilities. Ours-$City_p$ and Ours-$City_{GT}$ refer to results generated using input pseudo and GT semantic label, respectively.

## 6 CONCLUSION

In this paper, we propose a semantic-guided feature representation enhancement paradigm for self-supervised monocular trained depth estimation. By proposing an extra semantic branch, our method learns semantic contextual information for depth estimation. We further enhance the depth feature representations in local and global manner, by the proposed semantic-guided edge enhancement module as well as the semantic-guided multi-layer self-attention mechanism. Extensive experiments validate the superiority of our method. In the future research, we plan to involve explicit constraints with semantic information for better depth estimation.

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

# Appendix

## A  IMPLEMENTATION DETAILS

In this section, we show more implementation details of our method, including network architectures and the parameter selection strategies.

### A.1  NETWORK ARCHITECTURE

The network architectures are shown in Table 4, we describe the parameters of each layer in the semantic branch, depth branch as well as the proposed SEEM module. The architecture of a single attention module is similar to the pixel-wise self-attention used in other methods Fu et al. (2019); Johnston & Carneiro (2020), thus we do not illustrate it in detail.

Table 4: **Network architectures**. "in_chns" and "out_chns" denote the number of input and output channels. "resolution" refers to the downscaling factor with regard to the input image. "input" stands for the input of each layer, where "↑" means NN-based upsampling. In the Depth Decoder, "Attn" denotes the semantic-guided attentions and "SEEM" is the proposed SEEM module.

**Semantic Decoder**

| layer | kernel | stride | in_chns | out_chns | resolution | input | activation |
|---|---|---|---|---|---|---|---|
| S_upconv4 | 3 | 1 | 2048 | 256 | 32 | econv4 | ELU |
| S_iconv4 | 3 | 1 | 256, 1024 | 256 | 16 | ↑ S_upconv4, econv3 | ELU |
| S_upconv3 | 3 | 1 | 256 | 128 | 16 | S_iconv4 | ELU |
| S_iconv3 | 3 | 1 | 128, 512 | 128 | 8 | ↑ S_upconv3, econv2 | ELU |
| S_disp3 | 3 | 1 | 128 | 1 | 1 | S_iconv3 | Sigmoid |
| S_upconv2 | 3 | 1 | 128 | 64 | 8 | S_iconv3 | ELU |
| S_iconv2 | 3 | 1 | 64, 256 | 64 | 4 | ↑ S_upconv2, econv1 | ELU |
| S_disp2 | 3 | 1 | 64 | 1 | 1 | S_iconv2 | Sigmoid |
| S_upconv1 | 3 | 1 | 64 | 32 | 4 | S_iconv2 | ELU |
| S_iconv1 | 3 | 1 | 32, 64 | 32 | 2 | ↑ S_upconv1, econv0 | ELU |
| S_disp1 | 3 | 1 | 32 | 1 | 1 | S_iconv1 | Sigmoid |
| S_upconv0 | 3 | 1 | 32 | 16 | 2 | S_iconv1 | ELU |
| S_iconv0 | 3 | 1 | 16 | 16 | 1 | ↑ S_upconv0 | ELU |
| S_disp0 | 3 | 1 | 16 | 1 | 1 | S_iconv0 | Sigmoid |

**Depth Decoder**

| layer | kernel | stride | in_chns | out_chns | resolution | input | activation |
|---|---|---|---|---|---|---|---|
| D_upconv4 | 3 | 1 | 2048 | 256 | 32 | econv4 | ELU |
| aconv4 | 3 | 1 | 256, 256 | 256 | 32 | D_upconv4, S_upconv4 | ELU |
| Attn4 | 3 | 1 | 256 | 256 | 32 | aconv4 | / |
| D_iconv4 | 3 | 1 | 256, 1024 | 256 | 16 | ↑Attn4, econv3 | ELU |
| D_upconv3 | 3 | 1 | 256 | 128 | 16 | D_iconv4 | ELU |
| aconv3 | 3 | 1 | 128, 128 | 128 | 16 | D_upconv3, S_upconv3 | ELU |
| Attn3 | 3 | 1 | 128 | 128 | 16 | aconv3 | / |
| D_iconv3 | 3 | 1 | 128, 512 | 128 | 8 | ↑Attn3, econv2 | ELU |
| D_disp3 | 3 | 1 | 128 | 1 | 1 | D_iconv3 | Sigmoid |
| D_upconv2 | 3 | 1 | 128 | 64 | 8 | D_iconv3 | ELU |
| aconv2 | 3 | 1 | 64, 64 | 64 | 8 | D_upconv2, S_upconv2 | ELU |
| Attn2 | 3 | 1 | 64 | 64 | 8 | aconv2 | / |
| D_iconv2 | 3 | 1 | 64, 256 | 64 | 4 | ↑Attn2, econv1 | ELU |
| D_disp2 | 3 | 1 | 64 | 1 | 1 | D_iconv2 | Sigmoid |
| D_upconv1 | 3 | 1 | 64 | 32 | 4 | D_iconv2 | ELU |
| aconv1 | 3 | 1 | 32, 32 | 32 | 4 | D_upconv1, S_upconv1 | ELU |
| D_iconv1 | 3 | 1 | 32, 64 | 32 | 2 | ↑aconv1, econv0 | ELU |
| D_disp1 | 3 | 1 | 32 | 1 | 1 | D_iconv1 | Sigmoid |
| D_upconv0 | 3 | 1 | 32 | 16 | 2 | D_iconv1 | ELU |
| aconv0 | 3 | 1 | 16, 16 | 16 | 2 | D_upconv0, S_upconv0 | ELU |
| SEEM | 1 | 1 | 256,64 64,32,16 64,32,16 | 16 | 2 | enconv1, enconv0, ↑ aconv2, ↑ aconv1, ↑ aconv0, ↑ S_upconv2, ↑ S_upconv1, ↑ S_upconv0 | / |
| D_iconv0 | 3 | 1 | 16 | 16 | 1 | ↑ aconv0,SEEM | ELU |
| D_disp0 | 3 | 1 | 16 | 1 | 1 | D_iconv0 | Sigmoid |

**SEEM**

| layer | kernel | stride | in_chns | out_chns | input | activation |
|---|---|---|---|---|---|---|
| enc_conv | 1 | 1 | 256, 64 | 128 | enconv1, enconv0 | / |
| dec_conv | 1 | 1 | 64, 32, 16 | 112 | ↑ aconv2, ↑ aconv1,↑ aconv0, | / |
| sem_conv | 1 | 1 | 64, 32, 16 | 112 | ↑ S_upconv2, ↑ S_upconv1, ↑ S_upconv0 | / |
| conv1d_1 | 1 | 1 | 128, 112, 112 | 256 | enc_conv, dec_conv, sem_conv | ReLu |
| conv1d_2 | 1 | 1 | 256 | 256 | conv1d_1 | ReLu |
| conv1d_3 | 1 | 1 | 256 | 256 | conv1d_2 | ReLu |
| conv1d_4 | 1 | 1 | 256 | 16 | conv1d_3 | / |

## A.2 PARAMETER SELECTION FOR SEEM MODULE

The parameters used in SEEM module are $N$, $\mu$ and $c$. Inspired by previous works (Kirillov et al., 2020; Xian et al., 2020), we set the ratio of edge-based points (including edge points $\mathcal{S}_\mathcal{E}$ and disturb points $\mathcal{S}_\mathcal{D}$) to random points $\mathcal{S}_\mathcal{R}$ to be $4:1$. In this way, the ratio parameter $\mu$ is set to $0.4$. At the mean time, we randomly select over 10000 images from KITTI dataset to calculate the number of edge pixels, the mean value is $1233.01$. To make the number of sampled edge points $\mu N$ to be consistent with the real edge point number, we set $\mu N = 1200$. Thus, the resulting number of sampled points $N$ is set to 3000. Under this setting, we find the sampled edge-based points ($\mathcal{S}_\mathcal{E}$, $\mathcal{S}_\mathcal{D}$) are able to cover most of the distinct edges, and the random points $\mathcal{S}_\mathcal{R}$ are sampled uniformly on the whole image.

For the choice of parameter $c$, we compute the ratio of the edge points which are close to the GT border within range of $[-c, c]$. We use KITTI Semantic dataset (Geiger et al., 2012) for evaluation because it provides groundtruth KITTI semantic borders. As shown in Table 5, when $c$ is set to 3, the ratio is $87.75\%$. This means the latent sampling areas of $\mathcal{S}_\mathcal{D}$ has more than $87\%$ of overlappings with the real object borders. When $c$ is set to 5, the ratio is $92.20\%$, which has a relative small improvement ($4.45\%$) toward that of $c = 3$. At the mean time, since enlarging the range of the disturb points sampling will inevitably cut down the correlations between $\mathcal{S}_\mathcal{D}$ and $\mathcal{S}_\mathcal{E}$, we select $c = 3$ as the final choice.

Table 5: **Ratio of the edge points** that lie within range $[-c, c]$ of the GT border.

| c | 1 | 3 | 5 |
|---|---|---|---|
| Ratio (%) | 55.30 | 87.75 | 92.20 |

## A.3 CHOICE OF ATTENTION LAYERS

In our paper, we set the number of attention layers considering both performance and GPU capacity. Firstly, we train our model on images of size $[64, 224]$, with attentions on different decoding layers. The quantitative results are shown in Table 6. Compared to model with a single attention (row 1), models with multi-level attentions show better performances. At the mean time, the performances of different multi-level attentions are close to each other (row 2, 3, 4). Consider the self-attention mechanism involves tensor multiplications which consume considerable GPU resources, we select model with attentions on layer $(4, 3, 2)$ as the final choice.

Table 6: **Quantitative results with different number of attentions**. Results of several versions of proposed semantic-guided multi-level attentions on KITTI 2015 (Geiger et al., 2012).

| Layers with attentions | Abs Rel | Sq Rel | RMSE | RMSE$_{log}$ | $\delta \leq 1.25$ | $\delta \leq 1.25^2$ | $\delta \leq 1.25^3$ |
|---|---|---|---|---|---|---|---|
| 4 | 0.140 | 1.103 | 5.479 | 0.218 | 0.821 | 0.943 | 0.976 |
| 4,3,2 | 0.136 | 1.070 | 5.441 | 0.214 | 0.830 | 0.945 | 0.977 |
| 4,3,2,1 | 0.136 | 1.099 | 5.440 | 0.214 | 0.832 | 0.945 | 0.977 |
| 4,3,2,1,0 | 0.137 | 1.065 | 5.391 | 0.214 | 0.828 | 0.945 | 0.977 |

## B QUALITATIVE ABLATION ANALYSIS

We provide qualitative ablation analysis of our method. We show qualitative results of ablated versions of our method in Figure 7. Baseline with SEEM module produces sharper (green box in column 1) and more accurate (green box in column 4) depth borders, while the baseline with semantic-guided attention (SA) generates globally accurate and smooth depth predictions (blue boxes). When combined together, the advantages of two contributions are both kept in the final predictions, which yields better results. For example, the sharper borders of traffic sign on column 1 and more accurate border predictions on column 4 are kept in final results, while the final depth predictions of people are globally accurate, which are consistent with the results generated by semantic-guided attentions.

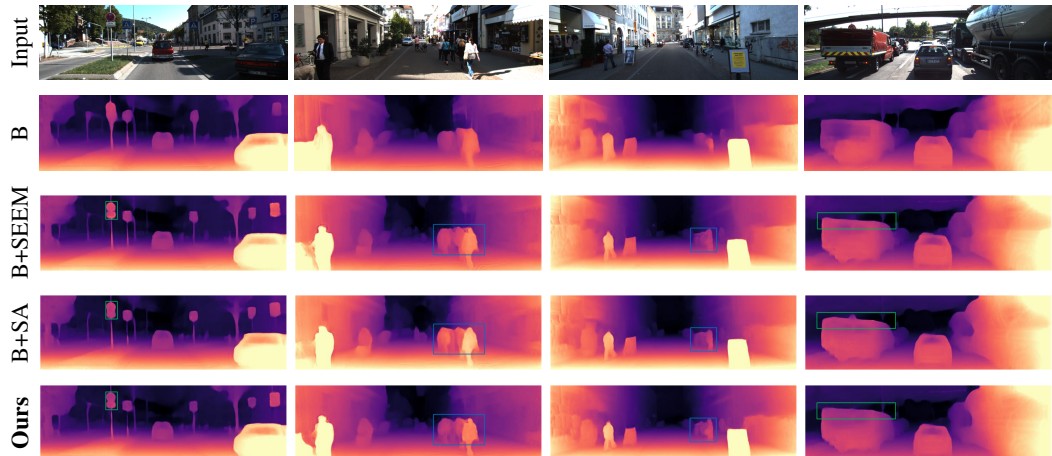

Figure 7: **Qualitative ablation results**. "B" denotes the baseline method. "B+SEEM" and "B+SA" are two ablated versions of our methods. As highlighted by the green and blue boxes, the two proposed contributions complement each other to produce predictions with the best accuracy.

## C   FURTHER DISCUSSION ON SEMANTICS

In this section, we provide further discussion on the semantic guidance used in our method. We give qualitative comparisons between pseudo labels generated with and without KITTI fine-tuning in Section C.1. Then, we show the foreground/background categories used for generating binary semantic map in Section C.2. Further more, we provide quantitative comparisons between our method and the state-of-the-art segmentation methods in Section C.3. Finally, in Section C.4, we evaluate two versions of our method that implemented with binary and full semantic branches, respectively.

### C.1   QUALITATIVE COMPARISON BETWEEN SEMANTIC MODELS WITH/WITHOUT KITTI FINE-TUNING.

We compare the semantic prediction accuracy between $M_{CSV+K}$ and $M_{CSV}$. The qualitative results of full/binary semantic predictions are shown in Figure 8. Row 3-4 are full semantic error map where the false predictions are marked with color. Row 5-6 are binary semantic error map, in which the false predictions are marked with gray. As shown in Figure 8, though the predictions between cross-domain trained $M_{CSV+K}$ and KITTI fine-tuned $M_{CSV+K}$ are obviously different in full semantic category, they are highly similar in terms of the binary category. The season is that the binary mask effectively alleviate the performance declines from full semantic labels, as illustrated in Section 5.4. For instance, though the "sidewalk" is falsely segmented as "road" by model trained on $M_{CS+V}$ in the third column, the erroneous area does not show in the binary error map because both "road" and "sidewalk" belong to the background category. This validates the advantage of using binary labels to alleviate the impact brought by cross-domain issues.

### C.2   GENERATION OF THE BINARY SEMANTIC MAP

Given the full semantic map, we generate the binary map by assigning the foreground objects to 1, and set the rest to 0. The detailed information is shown in Table 7.

Table 7: **Selection of foreground / background areas**.

| Foreground categories | traffic sign, person, rider, car, truck, bus, train, motorcycle, bicycle, traffic light |
|---|---|
| **Background categories** | road, sidewalk, building, wall, fence, pole, vegetation, terrain, sky |

### C.3   PREDICTION ACCURACY OF THE SEMANTIC BRANCH

We show the prediction accuracy of our semantic branch using groundtruth labels of KITTI semantic dataset (Geiger et al., 2012) in Table 8. Our model is trained on $M_{CSV}$-generated semantic labels in

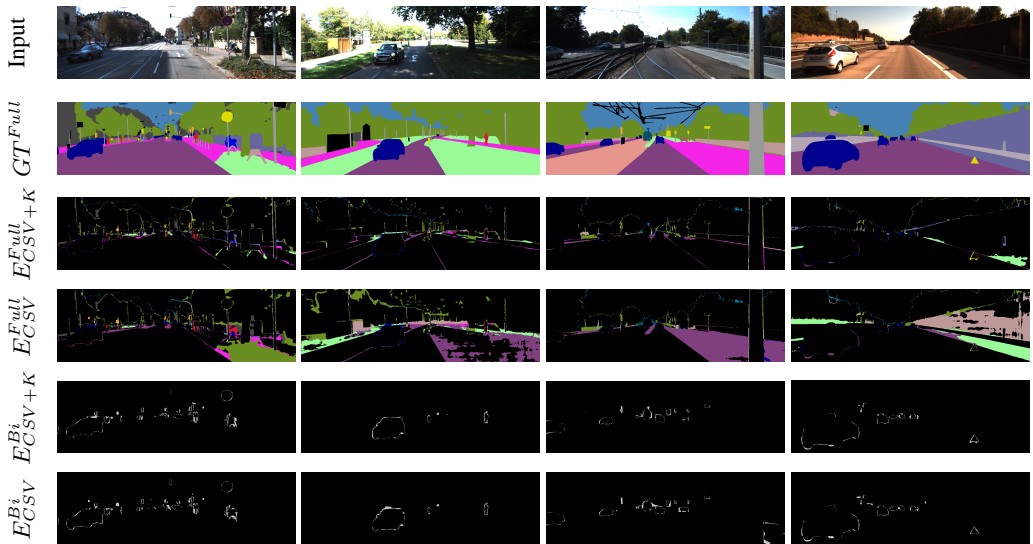

Figure 8: **Qualitative results of off-the-shelf semantic models**. "$E$" denotes the error map, "$CSV$" and "$CSV + K$" refer to results generated by model $M_{CSV}$ and $M_{CSV+K}$. "$Full$" means full semantic label with 19 categories, while "$Bi$" stands for the binary label containing only background/foreground categories.

order to rule out the influence of KITTI groundtruth. As shown on the last two columns, our semantic branch learns well from the $M_{CSV}$-generated pseudo semantic labels that their performances are very close to each other. At the mean time, our semantic branch produces competitive results toward state-of-the-art semantic segmentation methods (Li et al., 2020b;c;a) in terms of binary segmentation, which shows the effectiveness of our method in providing reliable foreground/background predictions.

Table 8: **Binary semantic segmentation accuracy**. We compare the binary prediction accuracy among our semantic branch, supervisory pseudo labels and other semantic segmentation methods.

| Method | Li et al. (2020b) | Li et al. (2020c) | Li et al. (2020a) | Ours | $M_{CSV}$ |
|---|---|---|---|---|---|
| mIoU$_{Bi}$ (%) | 89.23 | 92.43 | 92.94 | 92.59 | 93.54 |
| DICE (%) | 95.95 | 94.07 | 96.58 | 96.05 | 96.24 |

### C.4 COMPARISON BETWEEN THE IMPLEMENTATION OF BINARY/FULL SEMANTIC SEGMENTATION BRANCHES

To further validate the effectiveness of the proposed binary semantic branch, we conduct quantitative comparison between the semantic segmentation branches supervised by full/binary labels respectively. For the semantic branch trained by full semantic labels, we change the network to conduct pixel-wise predictions of 19 categories. The full semantic labels are pre-processed into one-hot vectors for supervision. Different from the network settings illustrated in the main text, the semantic features are directly concatenated with the depth feature without additional contributions. The results shown in Table 9 validate that there is no distinct performance difference between the networks supervised by full or binary semantic labels. Thus, we choose the binary semantic branch in our final settings as it well encodes the semantic contextual information with the preferable simplicity.

Table 9: **Quantitative comparison between different semantic segmentation branches**. "Full semantic label" refers to the semantic branch supervised by full category labels (19 categories), while "Binary semantic label" represents the model trained by the binary semantic label.

| Groundtruth type | Abs Rel | Sq Rel | RMSE | RMSE$_{log}$ | $\delta < 1.25$ | $\delta < 1.25^2$ | $\delta < 1.25^3$ |
|---|---|---|---|---|---|---|---|
| Full semantic label | 0.107 | 0.835 | 4.675 | 0.185 | 0.888 | 0.963 | 0.982 |
| Binary semantic label | 0.108 | 0.784 | 4.588 | 0.184 | 0.887 | 0.963 | 0.983 |

Table 10: **Quantitative results on KITTI improved ground truth**. "S" and "M" refer to self-supervision methods trained using stereo images and monocular images, respectively. Results are presented without any post-processing. The metrics marked in blue mean "lower is better", while these in red refer to "higher is better". Our method still produce comparable or better results compared to the state-of-the-art methods.

| Method | Training | Abs Rel | Sq Rel | RMSE | $RMSE_{log}$ | $\delta < 1.25$ | $\delta < 1.25^2$ | $\delta < 1.25^2$ |
|---|---|---|---|---|---|---|---|---|
| Godard et al. (2017) | S | 0.109 | 0.811 | 4.568 | 0.166 | 0.877 | 0.967 | 0.988 |
| Pillai et al. (2019) +pp | S | 0.090 | 0.542 | 3.967 | 0.144 | 0.901 | 0.976 | 0.993 |
| Godard et al. (2019) | S | 0.085 | 0.537 | 3.868 | 0.139 | 0.912 | 0.979 | 0.993 |
| Luo et al. (2018) | MS | 0.123 | 0.754 | 4.453 | 0.172 | 0.863 | 0.964 | 0.989 |
| Godard et al. (2019) | MS | 0.080 | 0.466 | 3.681 | 0.127 | 0.926 | 0.985 | 0.995 |
| Zhou et al. (2017) | M | 0.176 | 1.532 | 6.129 | 0.244 | 0.758 | 0.921 | 0.971 |
| Mahjourian et al. (2018) | M | 0.134 | 0.983 | 5.501 | 0.203 | 0.827 | 0.944 | 0.981 |
| Yin & Shi (2018) | M | 0.132 | 0.994 | 5.240 | 0.193 | 0.833 | 0.953 | 0.985 |
| Wang et al. (2018) | M | 0.126 | 0.866 | 4.932 | 0.185 | 0.851 | 0.958 | 0.986 |
| Ranjan et al. (2019) | M | 0.123 | 0.881 | 4.834 | 0.181 | 0.860 | 0.959 | 0.985 |
| EPC++ | M | 0.120 | 0.789 | 4.755 | 0.177 | 0.856 | 0.961 | 0.987 |
| Godard et al. (2019) | M | 0.090 | 0.545 | 3.942 | 0.137 | 0.914 | 0.983 | 0.995 |
| Guizilini et al. (2020a) | M | 0.078 | 0.420 | 3.485 | 0.121 | 0.931 | 0.986 | 0.996 |
| **Ours** | M | 0.081 | 0.437 | 3.635 | 0.124 | 0.927 | 0.987 | 0.996 |

## D  KITTI IMPROVED GROUND TRUTH RESULTS

We use KITTI (Geiger et al., 2012) with improved groundtruth (Uhrig et al., 2017) to further validate the effectiveness of our method. The test set is part of the Eigen split (Eigen et al., 2014), which accounts for 93% (652 / 691) of the original test frames. Quantitative results are shown in Table 10. Our method produces comparable or better results toward other state-of-the-art methods.

## E  ADDITIONAL QUALITATIVE RESULTS

We offer additional qualitative comparisons for depth on benchmark KITTI (Geiger et al., 2012) and Cityscapes (Cordts et al., 2016). All models are trained using only KITTI. As shown in Figure 9 (KITTI) and Figure 10 (Cityscapes), the proposed method achieves the best performance, that it produces sharp depth edges as well as detailed depth information from thin structures. Note that in Figure 10, the fourth and fifth column indicate depth maps generated with pseudo semantic guidance and groundtruth guidance, respectively. Results show that even tested with pseudo semantic guidance, our method still yields high quality depth predictions as the one guided with groundtruth.

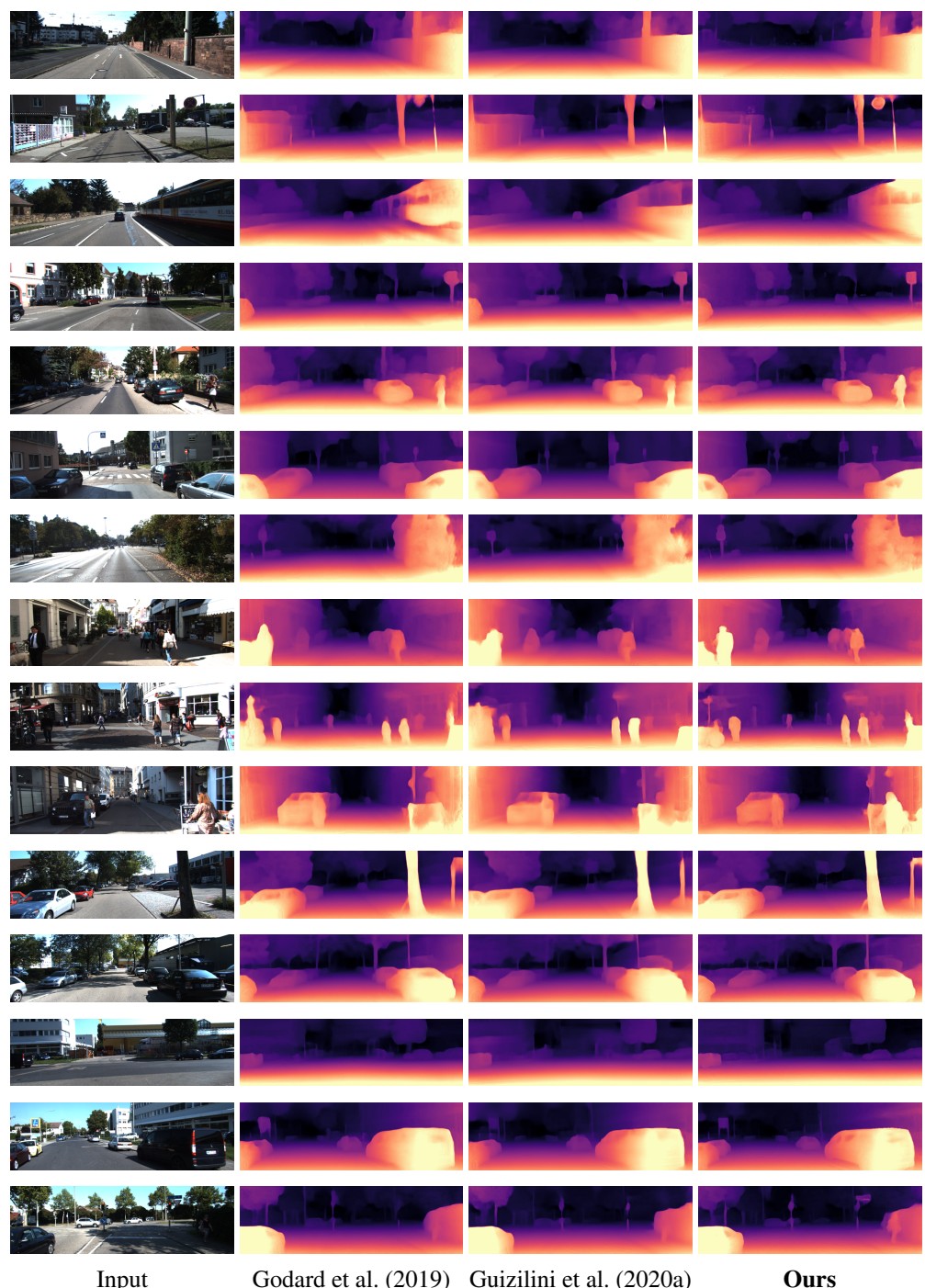

Input      Godard et al. (2019)    Guizilini et al. (2020a)     **Ours**

Figure 9: **Additional qualitative comparisons on KITTI dataset**. All models are trained on KITTI (Geiger et al., 2012) Eigen split (Eigen et al., 2014).

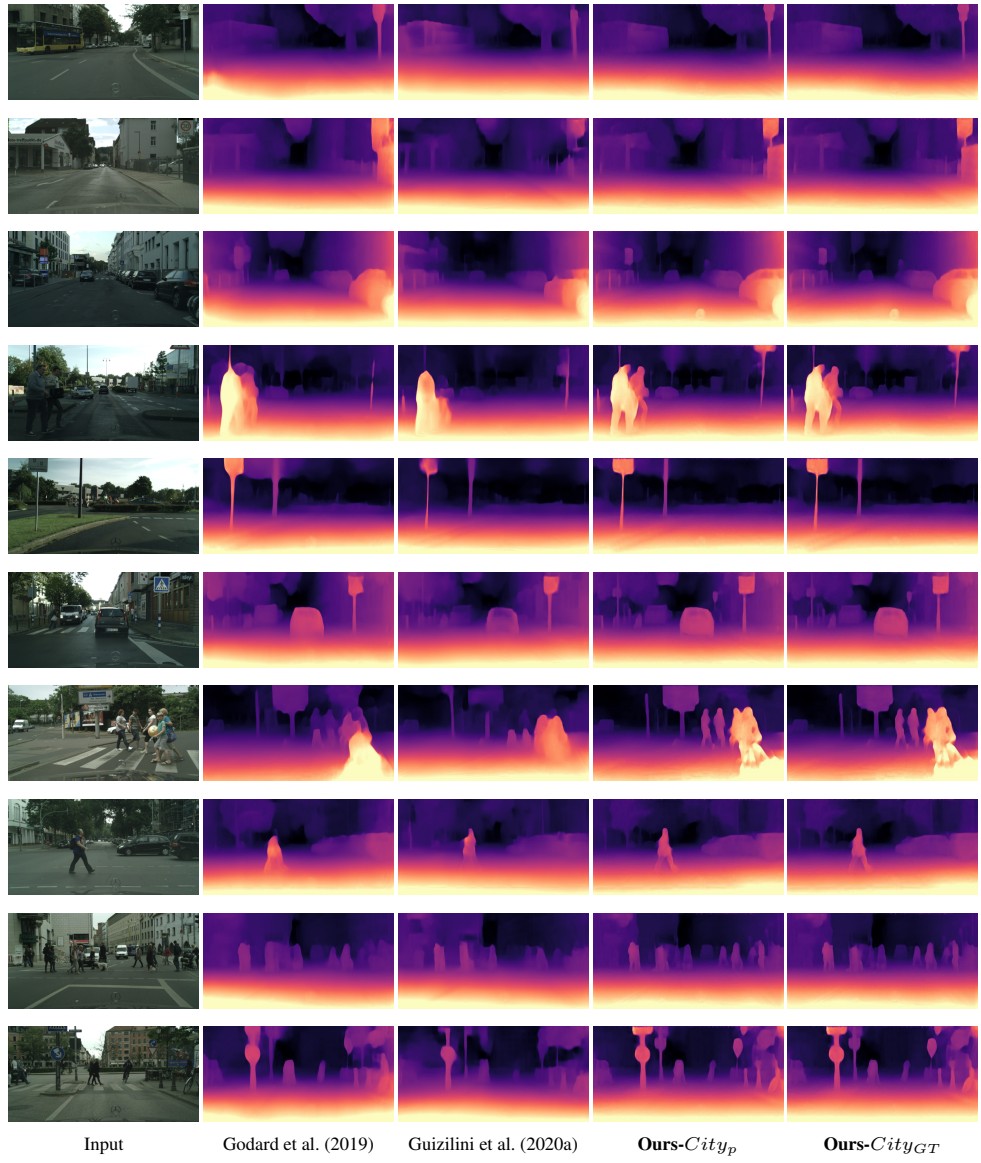

Figure 10: **Additional qualitative comparisons on Cityscapes**. All models are trained on KITTI (Geiger et al., 2012) Eigen split (Eigen et al., 2014). Ours-$City_p$ and Ours-$City_{GT}$ refer to results generated using input pseudo and GT semantic label, respectively.

