# OpenReview forum: "Semantic-Guided Representation Enhancement for Self-supervised Monocular Trained Depth Estimation"
_ICLR.cc/2021/Conference — Reject_

### Official Review · AnonReviewer1 · 2020-10-14
**This paper proposes to use semantic labels for improving the self-supervised depth estimation. The results show better depth estimation in object boundary areas. However, I have serious concerns about the semantic supervision, the claim of self-supervised learning, the trade-off between performance improvement and training data requirement, and the practicality of proposed method.**

**Rating:** 5
**Confidence:** 5

**Review:**

Strengths:

1. Writing is good. It is easy to read.
2. Although limited, the proposed term really improves the performance.
3. The qualitative results show better depth estimation in object boundary areas.



Weakness:

4. “How to generate semantic labels” is a very important problem in this paper. Simply referring to (Zhu et al., 2019) is not sufficient, and I suggest authors at least using 3-5 sentences to describe that, covering introduction for the input, output, and methodology.

5. After reading (Zhu et al., 2019), I find that the ground-truth semantic labels are still required, and their method can be regarded as a data augmentation. Then I have serious concerns about semantic supervisions:
a. The ground truth for semantics requires human labelling, while the depth supervision needs LiDAR or Kinect-like sensors. In my opinion, the former one is more expensive than the latter, so the requirement of training the proposed model is higher than existing supervised methods.
b. By comparing the results, I find that the proposed method is not as good as supervised methods or self-supervised methods that use stereo pairs for training.
c. Due to the requirement for ground truth semantics, the claim of self-supervision is not convincing.
d. It loses the advantage of self-supervised depth estimation. For example, existing depth CNN (Gordon et, al, iccv 2019) can be trained on a wide range of YouTube videos, where no semantic labels are provided. Although the pre-trained segmentation network could be used, their generalization is not validated.

6. The proposed method leads to very limited performance improvement, e.g., final results (absrel=0.105) vs baseline results (absrel=0.110) as shown in the Table 2, while it requires additional semantic labels for training. Considering the trade-off between performance improvement and requirement for training data, I have serious concerns about the contribution of this paper.

7. Object boundary is hard to estimate in all dense prediction tasks. This paper improves that in depth estimation by using the well-labeled semantic maps. However, in practice, obtaining accurate semantic labels is difficult, and this is also a challlenging problem for semantic segmentation.

8. The importance of depth sharpness on object boundary should be more clearly discussed, and the trade-off between such improvement and disadvantages caused by proposed solutions (as I mentioned above) should be discussed.

post-rebuttal: Authors pay much effort on addressing issues that I mentioned, and I appreciate that very much. My issues can be partially resoved, but the main issue is regarding the high-level thinking on semantic supervisions, which is impossible to be fully addressed in a rebuttal. Besides, I agree with AC that (a) the improvement is very limited and (b) semantic labels are hard to obtain. Overall, I raise my rating to 5 (not so good, but could be accepted), because I really appreciate authors' effort.

---

> ### Author Response · Authors · 2020-11-24
> **Response to Reviewer 1（ Part 1）**
>
> We really appreciate the reviewer's thoughtful comments on our manuscript, since the low-cost semantic supervision that the reviewer have concerns about is the prerequisite of our work. We address all concerns of the reviewer that we experimentally justify the feasibility of semantic supervision under self-supervision settings. We also discuss the contribution and practicability of our method in this context.
>
>
>
> **Q1:** Describe the generation of semantic labels, covering introduction for the input, output, and methodology.
>
> **A1:** The input semantic pseudo labels can be obtained by pre-trained semantic models. In this paper, we use Zhu's [1] model which is trained on labeled dataset (not necessarily on KITTI). To further validate the feasibility of using this semantic model at comparatively low cost, we prepare two versions of Zhu's model ($M_{CSV+K}$ and $M_{CSV}$) that trained with different settings for comparison.
>
> The main semantic model $M_{CSV+K}$ that we use for our network is pre-trained on Cityscapes and Mapillary Vistas dataset [2], and fine-tuned with KITTI Semantic (contains 200 training images) using 10-split cross validation strategy. Compared with the large quantity of KITTI raw data used for self-supervision (more than 39,000 training items), the required labeled images only account for a very small proportion, which indicate a relatively lower cost. At the mean time, to cover the scenario that no groundtruth KITTI label is provided,  we provide another semantic model $M_{CSV}$, which is pre-trained only on Cityscapes and Mapillary Vistas dataset [2], without involving any images from KITTI.
>
> During semantic dataset generation, the raw KITTI data is fed into the pre-trained semantic model to get the fully segmented images. Then, we get the binary semantic labels by assigning the traffic signs, different type of vehicles, and people, *etc.* as foreground. The label generation step is introduced in Section 5.1 of the updated paper.

---

> ### Author Response · Authors · 2020-11-24
> **Response to Reviewer 1（ Part 2）**
>
> **Q2:** Concerns about semantic supervisions:
>
> (1) Expensive semantic groundtruth labeling for training the off-the-shelf semantic model.
>
> (2) Proposed method is not as good as depth supervised or stereo-trained methods.
>
> (3) The claim of self-supervision is not convincing.
>
> (4) It loses the advantage of self-supervised depth estimation.
>
> **A2:** Among all these concerns, addressing the first one is of great significance to address others.
>
> (1) We agree with the reviewer that the semantic model should be trained with groundtruth semantic labels. However, what we intend to emphasize is that the semantic model can be pre-trained with other datasets, with the minor or even no fine-tunning step on current dataset, which indicates a much lower cost to achieve the goal.
>
> **Firstly**, the main semantic model $M_{CSV+K}$ that we use is not trained with large quantity of labeled images from the KITTI raw dataset. Instead, the model is fine-tuned with KITTI Semantic which only contains 200 labeled images. Since $M_{CSV+K}$ is able to generate appropriate results across more than 39,000 images with only 200 labeled images as groundtruth guidance, the cost seems not to be so expensive as it is initially considered.
>
> **Secondly**, consider the scenarios where no labeled image is provided for current dataset, we provide model $M_{CSV}$ which is trained without KITTI fine-tuning. We compare the results of pseudo semantic labels generated by  $M_{CSV+K}$ and $M_{CSV}$, the quantitative and qualitative results are shown in Table 3 and Figure 8 respectively. We find an interesting phenomenon that although $M_{CSV+K}$'s performance (79.90%) is obviously better than $M_{CSV}$'s (67.73%) in terms of $mIoU_{Full}$, both methods’ performances of $mIoU_{Bi}$ are very close (94.74% VS 93.54%). We attribute it to the reason that the false segmentations often occur between categories solely belong to the background (or the foreground). Thus, the binary mask effectively rules out the performance declines from full semantic labels. This indicates a general advantage which still hold for other models trained with other datasets. At the mean time, quantitative results of Table 3 show that our model trained on *$M_{CSV}$* semantic labels generates comparable results toward that trained on$M_{CSV+K}$ labels. This validates that the network performance will not be greatly influenced when no groundtruth label is provided for the on-the-shelf semantic model, which further prove the feasibility of our method. We update this part in Section 5.4 and Appendix C.1 of the revised paper.
>
> **Thirdly**, using the pre-trained semantic model to guide the depth estimation is broadly used in current methods [6, 7, 8, 9]. Among these methods, [6] uses a pre-trained segmentation network on Cityscape for feature-level guidance, while [7] uses Zhu's [1] model with the same settings as $M_{CSV+K}$ to generate semantic pseudo labels. We strike a balance between these methods that we provide both the fine-tuned model and the pre-trained model to validate the effectiveness of our method.
>
> After the detailed analysis, we address the last three concerns. Consider the semantic supervision can be achieved by fine-tuning on a small subset of labels or directly using pre-trained models from other datasets, our method inherits the merits of monocular self-supervision.
>
> (2) Compared with supervised methods and stereo self-supervised methods, our method are not restricted by depth labels or stereo inputs for depth training, and provides genuine improvements towards prevailing methods [3, 4, 5] of the other two categories.
>
> (3) Due to the low cost to get semantic pseudo labels as we illustrate above, we think the self-supervision can also be achieved under this setting. Moreover, the previous methods [7, 8] also show the practicability of achieving self-supervised depth estimation with pre-trained or fined-tuned semantic models.
>
> (4) For the other tasks such as learning depth from Youtube videos, one can select the semantic segmentation model which is pre-trained on datasets related to the task scenarios, to further improve the quality of pseudo labels. We agree with the reviewer that the performance of these tasks should further be validated. But in this paper, we mainly focus on testing and validating on the driving scenarios, following the practices of other self-supervised methods. Within this scope, we validate the effectiveness of using the semantic models pre-trained on other datasets. For the performance of other tasks for self-supervised depth, we would like to conduct further research in the future work.

---

> ### Author Response · Authors · 2020-11-24
> **Response to Reviewer 1（ Part 3）**
>
> **Q3:** The proposed method leads to limited performance improvement, while it requires additional semantic labels for training.
>
> **A3:** We illustrate the feasibility for semantic training in **A2** that we do not address it in detail here. For the performance improvement, we would like to illustrate the effectiveness of our method in threefold. Firstly, our method leverages the semantic information as additional guidance. So when new architectures or pipelines are introduced, our method will bring continuous improvements upon the success of others. Secondly, our method shows its superiority in estimating depth on borders or thin structures, which brings the advantages that beyond the absolute metric itself. We will further discuss it in **A5**. Thirdly, the performance improvement of our method is about 5%, which is consistent with the typical year-over-year performance improvement, as illustrated by [6] in their discussion pages.
>
>
>
> **Q4:** Object boundary is hard to estimate in all dense prediction tasks, and this is also a challenging problem for semantic segmentation.
>
> **A4:** We agree with the reviewer that the object boundary is hard to estimate in all dense prediction tasks including semantic segmentation. But as validated by [7], semantic segmentation even trained with limited groundtruth can offer more accurate border that of any image based depth estimation. In this situation, even though the semantic segmentation can not perfectly align the real border, it still offer better guidance for depth estimation.
>
>
>
> **Q5:** The importance of depth sharpness on object boundary should be discussed, and the trade-off between such improvement and disadvantages should be discussed.
>
> **A5:** Improving the quality of the depth borders not only contributes to the improvement of quantitative performance, but also brings important information indicating the relative positional relationship between objects. The blurry depth boundaries will lead to the confusion between the foreground and background, which brings negative impacts for the subsequent tasks such as autonomous driving. As shown in the last row Figure 9, the blurry depth contour of the right person makes the one hard to be distinguished from the background. Our method addresses this issue that it produces clear depth borders via the proposed local and global feature representation enhancement schemes.
>
> Since the distinct depth borders are of notable importance, we can get well estimated depth borders by leveraging the pre-trained semantic model. In this situation, the use of pre-trained segmentation model does not yield obvious disadvantages, compared with the benefits it  brings.
>
>
>
>
>
> [1] Zhu Y, Sapra K, Reda F A, et al. Improving semantic segmentation via video propagation and label relaxation[C]//Proceedings of the IEEE Conference on Computer Vision and Pattern Recognition. 2019: 8856-8865.
>
> [2] Neuhold G, Ollmann T, Rota Bulo S, et al. The mapillary vistas dataset for semantic understanding of street scenes[C]//Proceedings of the IEEE International Conference on Computer Vision. 2017: 4990-4999.
>
> [3] Watson J, Firman M, Brostow G J, et al. Self-supervised monocular depth hints[C]//Proceedings of the IEEE International Conference on Computer Vision. 2019: 2162-2171.
>
> [4] Godard C, Mac Aodha O, Firman M, et al. Digging into self-supervised monocular depth estimation[C]//Proceedings of the IEEE international conference on computer vision. 2019: 3828-3838.
>
> [5] Kuznietsov Y, Stuckler J, Leibe B. Semi-supervised deep learning for monocular depth map prediction[C]//Proceedings of the IEEE conference on computer vision and pattern recognition. 2017: 6647-6655.
>
> [6] Guizilini V, Hou R, Li J, et al. Semantically-Guided Representation Learning for Self-Supervised Monocular Depth[J]. arXiv preprint arXiv:2002.12319, 2020.
>
> [7] Zhu S, Brazil G, Liu X. The Edge of Depth: Explicit Constraints between Segmentation and Depth[C]//Proceedings of the IEEE/CVF Conference on Computer Vision and Pattern Recognition. 2020: 13116-13125.
>
> [8] Chen P Y, Liu A H, Liu Y C, et al. Towards scene understanding: Unsupervised monocular depth estimation with semantic-aware representation[C]//Proceedings of the IEEE Conference on Computer Vision and Pattern Recognition. 2019: 2624-2632.
>
> [9] Wang L, Zhang J, Wang O, et al. SDC-Depth: Semantic Divide-and-Conquer Network for Monocular Depth Estimation[C]//Proceedings of the IEEE/CVF Conference on Computer Vision and Pattern Recognition. 2020: 541-550.

---

### Official Review · AnonReviewer2 · 2020-10-27
**Official Blind Review #2**

**Rating:** 6
**Confidence:** 5

**Review:**

Summary:

This paper proposed a novel framework to improve self-supervised monocular depth estimation leveraging semantic features at local and global level. The proposed framework includes a semantic-guided edge enhancement module to extract and enhance point-based features around semantic boundaries. The proposed framework also incorporates feature fusion through a self-attention module at different feature levels for global fusion. Evaluation on KITTI datasets compared with recent state-of-the-arts are provided. The proposed method outperformed other monocular depth estimation methods including the ones that also leverage semantic information.

--------------------------------------------------------------------------------------------

Pros:
* The paper is well formatted and easy to read.
* The idea of semantic guidance both locally and globally is novel and interesting.
* The new framework performs the previous state-of-the-arts under the same network settings.

Cons:
* The cityscapes experiment provides very limited contribution to the paper as only qualitative results are provided.
* The prediction accuracy of semantic segmentation branches is missing. Even Though it is trained with pseudo labels, it is still important to provide this result in order to justify that the semantic branch is providing reasonable predictions.
* There is some other confusing description in the experiment section that raises concerns:
  - The paper keeps mentioning that the semantic pseudo labels are generated from off-the-shield algorithms which require no groundtruth. This argument is confusing as the pretrained model from reference work did use ground truth.
  -The authors mentioned semantic labels for the test set of KITTI and Cityscapes. Why do you need to use these labels for testing when no semantic quantitative result is provided? Please provide more details there. Or one might assume the label is needed during network inference.
  - The author mentioned that the final model is selected via validation set. Does that mean the reported number is not from a model converged with the training protocol? Or is it selected from multiple runs with random seed? Please provide more details there.

----------------------------------------------------------
Other detail comments:

- In related work, self-supervised depth estimation, I think it would be good to mention the effort of network architecture improvement for depth estimation (E.g. Guizilini 2019a). Then emphasize that the experiment comparison rules out the impact of different network architecture in either related work or experiment section, given that the main result in Guizelini 2019b using another network is not compared in the table.
- Since both depth branches and semantic branches are trained together, how do you weigh the losses? It is a simple sum up or weighted? Please provide the detail for reproducibility.
- In figure 4. Is the semantic guidance map the groundtruth label or predicted maps?
- The author mentioned in Appendix C that binary semantic categorization is used. Please provide details on what classes are considered as foreground.
- The description “insufficient depth representation” is a bit confusing. The solution to “insufficient” should be “providing more”, while the paper is improving and augmenting the representation. I would recommend the author consider changing the description here.
- It would be nice to include some qualitative results along with the ablation analysis.
--------------------------------------------------------
post-rebuttal:
I thank the authors for their clarification and ablation analysis. Most of my concerns are resolved. I will keep my original score and I think the paper could be accepted.

---

> ### Author Response · Authors · 2020-11-24
> **Response to Reviewer 2（ Part 1）**
>
> We thank the reviewer for the positive review and the helpful comments to further improve our paper. We address all comments raised by the reviewer, and we have updated the corresponding parts in the revised paper.
>
>
>
> **Q1:** The Cityscapes experiment provides very limited contribution to the paper as only qualitative results are provided.
>
> **A1:** Thank you for the advice. We cut down the number of shown Cityscapes results in the main body of the paper. At the mean time, we add an column to show extra results of our model, which are generated with Cityscapes pseudo labels as guidance (the previous results are guided by Cityscapes groundtruth labels). It validates that our method consistently conducts high quality predictions regardless of the type of input semantic guidance.
>
>
>
> **Q2:** The prediction accuracy of semantic segmentation branches is missing. It is still important to provide this result in order to justify that the semantic branch is providing reasonable predictions.
>
> **A2:** We provide the binary semantic segmentation accuracy of our method, and compare it with state-of-the-art segmentation methods [1, 2, 3] in Table 8 of the Appendix.  We test these models via KITTI Semantics which has groundtruth labels. For fair comparison, we train our model on pseudo labels generated by Zhu's [4] model $M_{CSV}$, which is pre-trained on Cityscapes and Vistas dataset without fine-tunning on KITTI. As shown on the last two columns of Table 8, our semantic branch learns well from the $M_{CSV}$-generated pseudo semantic labels that their performances are very close to each other. At the mean time, our semantic branch produces competitive or even better results toward state-of-the-art semantic segmentation methods in terms of binary segmentation, which shows the effectiveness of our method in providing reliable foreground/background predictions. We add this part in Appendix C.3 of the updated paper.
>
>
>
> **Q3:** Some other confusing description in the experiment section. (1) The pretrained semantic model from reference work did use ground truth. (2) Why need to use semantic labels for testing when no semantic quantitative result is provided? (3) Please provide more details there for the selection of the final model.
>
> **A3:** (1) In the previous paper, the illustration here is indeed confusing. The off-the-shelf segmentation method is indeed trained with labeled images. However, what we intend to emphasize here is that the semantic model can be well pre-trained from other datasets, and few or no KITTI groundtruth labels  are required for the semantic model. To validate this claim, we set two versions of Zhu's models. $M_{CSV+K}$ is the semantic model fine-tuned with 200 KITTI labels, and $M_{CSV}$ is the model without any KITTI fine-tunning. We find the two models show similar performances in terms of binary segmentation (shown in Table 3 and Figure 8). At the mean time, the performances of our model under the two semantic datasets are also close to each other, which validates that our method can use the semantic model which is trained using other datasets, without involving KITTI semantic groundtruth. Detailed analysis can be found on Section 5.4 and Appendix C.1 of the updated paper.
>
> (2) During the testing phase, the semantic maps are used as input and they can either be the groundtruth labels or the pre-computed pseudo labels. In our paper, we test KITTI with the pre-computed pseudo labels. For the Cityscapes dataset, we validate in Section 5.5 of the updated paper that either pseudo labels or groundtruth labels can be used as guidance for testing. We also report the performance of the semantic branch in Table 8, but we do not use the semantic maps here to generate depth maps.
>
> (3) During network training, we judge the convergence of the network by both loss curve and the performance on validation dataset. When the loss curve gets smooth, the network performance becomes stable, we select the model in that situation as the final choice.

---

> ### Author Response · Authors · 2020-11-24
> **Response to Reviewer 2（ Part 2）**
>
> **Q4:** Other detail comments:
>
> (1) Emphasize that the experiment comparison rules out the impact of different network architecture.
>
> (2) How do you weigh the losses of depth branches and semantic branches?
>
> (3) In figure 4. Is the semantic guidance map the groundtruth label or predicted maps?
>
> (4) Please provide details on what classes are considered as foreground.
>
> (5) The description “insufficient depth representation” is a bit confusing.
>
> (6) It would be nice to include some qualitative results along with the ablation analysis.
>
> **A4:**
>
> (1) In the updated paper, we mention the contributions of Guizilini et al. (2020b) in novel architecture design. And we emphasize both in related work and experimental section that we do not consider the performance improvement brought by better network architectures when conducting quantitative comparisons. At the meantime, as suggested by Reviewer #3, we also report the results of Guizilini et al. (2020b) on PackNet backbone for better reference.
>
> (2) We directly sum the losses of depth and semantic branch together without assigning the weight. And we add the illustration of the final loss in Section 4.4 of the updated paper.
>
> (3)  In Figure 4, the semantic guidance maps for KITTI testing are the predicted pseudo labels. We also compare the results guided by either pseudo labels and groundtruth labels on Cityscapes.  As shown in Figure 6 and Figure 10, there is no obvious difference.
>
> (4) We show the detailed foreground and background categories in Table 7 of the Appendix. The category "traffic sign", "person", "rider", "car", "truck", "bus", "train", "motorcycle", "bicycle" and "traffic light" are regarded as the foreground.
>
> (5) The illustration is a bit confusing here, thank you for the reminder. We update the description as "limited depth representation ability" in the abstract and introduction.
>
> (6) That's a great idea to show the qualitative ablation analysis. We add this part on Section B of the Appendix. Qualitative results can be found in Figure 7. We find that baseline with SEEM module produces sharper (green box in column 1) and more accurate (green box in column 4) depth borders, while the baseline with semantic-guided attention (SA) generates globally accurate and smooth depth predictions (blue boxes). When combined together, the advantages of two contributions are both kept in the final predictions, which yields better results. For example, the sharper borders of traffic sign on column 1 and more accurate border predictions on column 4 are kept in final results, while the final depth predictions of people are globally accurate, which are consistent with the results generated by semantic-guided attentions.
>
>
>
>
>
> [1] Li X, You A, Zhu Z, et al. Semantic Flow for Fast and Accurate Scene Parsing[C]//European Conference on Computer Vision. Springer, Cham, 2020: 775-793.
>
> [2] Li X, Zhao H, Han L, et al. Gated Fully Fusion for Semantic Segmentation[C]//Proceedings of the AAAI Conference on Artificial Intelligence. 2020, 34(07): 11418-11425.
>
> [3] Li X, Li X, Zhang L, et al. Improving Semantic Segmentation via Decoupled Body and Edge Supervision[J]. arXiv preprint arXiv:2007.10035, 2020.
>
> [4] Zhu Y, Sapra K, Reda F A, et al. Improving semantic segmentation via video propagation and label relaxation[C]//Proceedings of the IEEE Conference on Computer Vision and Pattern Recognition. 2019: 8856-8865.

---

### Official Review · AnonReviewer3 · 2020-10-28
**Official Blind Review #3**

**Rating:** 7
**Confidence:** 4

**Review:**

The authors tackle the problem of self-supervised depth estimation and particularly address the issue of poor depth estimation on object boundaries. The introduction motivates the problem well, and the related work covers most of the relevant papers. The authors propose two main modifications allowing them to leverage an off-the-shelf semantic segmentation network: SEEM (semantic guided edge enhancement module) and a multi-level self-attention mechanism that fuses depth and semantic features at different levels. The combination of these contributions along with an off-the-shelf semantic segmentation network achieves impressive results, especially on the  boundaries of objects.

Overall the paper is well written and the method clearly explained; the 2 diagrams in Figures 1 and 2 are very well done and explain the workings of the system quite well. The authors ablate their contributions clearly showing their respective impact on the overall performance. The evaluation on KITTI shows that the method compares favorably with related work. The presentation could be improved by addressing the points mentioned below, and specifically: (i) it would be interesting to see how the quality of the off-the-shelf semantic segmentation network affects the depth results (see below) and (ii) it would be nice to see a per semantic class evaluation (similar to Fig 4 in  (Guizilini et al., 2020b)) showing specific improvements after using the proposed modifications.

Comments/suggestions for improvement:

* Does the quality of the off-the-shelf segmentation affect the result? There is an implicit assumption here that the pretrained, off-the-shelf network transfers well to the current domain. It would be interesting to explore how this affects depth estimation performance.
* What data is the pretrained semantic segmentation network of Zhu et al. (2019) trained on? Could the authors report the performance of the network on the original training dataset, for reference? Guizilini et al. (2020b) report that their pre-trained semantic segmentation network achieves an mIoU of 75% on the Cityscapes val set; how does that compare to the network of Zhu et al. (2019) used by the authors?
* Table 1 - the numbers reported for Guizilini et al. (2020b) don’t seem to include the PackNet numbers reported in that paper (i.e. the abs_rel reported for 640x192 seems to be 0.102); while that network architecture is different, I would argue it’s still relevant as a comparison with state-of-the-art methods
* Table 3 in the supplementary is missing the Guizilini et al. (2020a) numbers on the KITTI improved ground truth.
* The paper is missing some details regarding the network architectures used - the authors should provide this information if possible, either in the main text or in the supplementary
* It would also be interesting to see the performance of the method with using full resolution images - would using semantic segmentation further help improve results in that setting?

Minor comments:
* Table 1 caption “D” denotes depth supervision - since there is no method using Depth supervision in the table this can be removed
* It would be nice if the authors could add F_E, F_D and F_S on Fig 1 to make it easier to follow the description in Sec 3.2
* Setting i \in [2,4] means using the self attention twice? Fig 1 has 3 self-attention blocks.

Post rebuttal: I thank the authors for their detailed response and paper revision. My concerns have been addressed, and I particularly appreciate the experiments presented Table 3, Figure 5 and Section 5.4. I am happy to maintain my rating and recommend acceptance.

---

> ### Author Response · Authors · 2020-11-24
> **Response to Reviewer 3（ Part 1）**
>
> We are really grateful for the reviewer's appreciation of our method and the valuable suggestions to further improve our paper. We address all the comments posted by the reviewer and update them in the revised paper.
>
> **Q1:** It would be nice to see a per semantic class evaluation showing specific improvements after using the proposed modifications.
>
> **A1:** That's a great idea. We provide category-level improvement analysis on Section 5.4. We show the performance improvements on each semantic category toward the baseline method [2]. The results are shown in Figure 5. Consider there is no groundtruth semantic labels for KITTI raw, we use the pseudo label to find semantic categories. As shown in Figure 5, except for category "Motorcycle" (performances are close with a very small margin of 0.00016), our method improves consistently across all other categories, including foreground category "traffic sign", "person", "car'' and background category "sky'', "building'' and "vegetation''. That validates that our method not only learns good representations of border areas, but also benefits from the global semantic contextual information provided by semantic guidance.
>
>
>
> **Q2:** What data is the pretrained semantic segmentation network of Zhu et al. (2019) trained on? Could the authors report the performance of the network on the original training dataset, for reference?
>
> **A2:** In our paper, the main semantic model $M_{CSV+K}$ that we use is pre-trained on Cityscapes plus Mapillary Vistas [1] datasets, and fine-tuned with 10-split cross validation on KITTI Semantics which contains 200 labeled images. The model achieves mIoU of 80% on the Cityscapes validation set. Although the number of labeled KITTI images used for training $M_{CSV+K}$ are very small compared with the KITTI training items for self-supervision (more than 39,000), groundtruth information is still involved. Consider this situation, in the updated paper we provide another version of Zhu's model $M_{CSV}$, which is a pre-trained model without KITTI fine-tunning.  We then compare our method trained using pseudo labels generated by $M_{CSV+K}$ and $M_{CSV}$ respectively, and the results show similar performances. The detailed analysis of this can be found in Section 5.4 and Appendix C.1 of the updated paper.
>
>
>
> **Q3:** Does the quality of the off-the-shelf segmentation affect the result?
>
> **A3:** It's very important to see how the performance of our method changes when trained using pseudo semantic labels with different qualities. To this end, we set two versions of Zhu's models. $M_{CSV+K}$ is the semantic model fine-tuned with 200 KITTI labels, and $M_{CSV}$ is the model without any KITTI fine-tunning. We first compare the quality of semantic labels on KITTI. As shown in Table 3 and Figure 8 in the updated paper, we find that the full semantic segmentation quality of $M_{CSV+K}$ is obviously better than that of $M_{CSV}$ (79.90% vs 67.73%). However, the binary semantic segmentation qualities of the two models are very close (94.74% vs 93.54%). In other words, the poorly segmented areas of different segmentation methods will be effectively removed by leveraging the binary semantic mask, which validates a general advantage for using binary semantic labels. We give detailed illustrations to this advantage on Section 5.4 and Appendix C.1. We also find in Table 3 that the performances of our model under the two semantic datasets are close to each other, which validates that the difference semantic segmentation results with different qualities lead small impact to our method.
>
>
>
> **Q4:** The numbers reported for Guizilini et al. (2020b) don’t seem to include the PackNet numbers reported in that paper, I would argue it’s still relevant as a comparison with state-of-the-art methods.
>
> **A4:** We add the quantitative results of Guizilini et al. (2020b) with PackNet architecture in Table 1 of the updated paper for better reference. And as suggested by Reviewer #2, we also illustrate in the paper that during experimental comparison, we did not consider the  performance improvements contributed by the novel architectures.
>
>
>
> **Q5:** Table 3 in the supplementary is missing the Guizilini et al. (2020a) numbers on the KITTI improved ground truth.
>
> **A5:** Thanks for the reminder, we added the numbers of Guizilini et al. (2020a) in Table 10 of the updated paper.

---

> ### Author Response · Authors · 2020-11-24
> **Response to Reviewer 3（ Part 2）**
>
> **Q6:** The paper is missing some details regarding the network architectures used - the authors should provide this information if possible, either in the main text or in the supplementary.
>
> **A6:** We add the network details in Table 4 of the Appendix, including architectures of the semantic branch, depth decoder as well as the proposed SEEM module. The architecture of attention module is similar to the pixel-wise self-attention used in other methods [3, 4], thus we do not illustrate here.
>
>
>
> **Q7:** It would also be interesting to see the performance of the method with using full resolution images - would using semantic segmentation further help improve results in that setting?
>
> **A7:**  Since higher resolution images requires more GPU memory, due to limited time and resources, we do not extend our model to run on multi-GPUs for full resolution images. However, we heuristically train our model on images with relative smaller resolution ($128 \times 416$), and compare the results with our model trained on current resolution ($192 \times 640$). The results are shown below, we can see that the improvement of resolution brings significant improvements under our network architecture, which indicates the potential of our network to leverage rich image and contextual information from high resolution training data.
>
> | Resolution       |  AbsRel   |   SqRel   |   RMSE    |  RMSElog  | $\delta<1.25$ | $\delta<1.25^{2}$ | $\delta<1.25^{3}$ |
> | ---------------- | :-------: | :-------: | :-------: | :-------: | :-----------: | :---------------: | :---------------: |
> | $128 \times 416$ |   0.117   |   0.928   |   4.860   |   0.192   |     0.869     |       0.959       |       0.981       |
> | $192 \times 640$ | **0.105** | **0.764** | **4.550** | **0.181** |   **0.891**   |     **0.965**     |     **0.983**     |
>
>
>
> **Q8:** Minor comments: (1) Table 1 caption "D" denotes depth supervision can be removed, (2) add F_E, F_D and F_S on Fig 1 to make it easier to follow, (3) Setting i \in [2,4] means using the self attention twice? Fig 1 has 3 self-attention blocks.
>
> **A8:**
>
> (1) We made a mistake here, and we remove this sentence in Table 1 of the updated paper.
>
> (2) We add $F_{E}$, $F_{D}$ and $F_{S}$ on Figure 1, to denote the layers used for point feature extraction.
>
> (3) We use the self-attention 3 times on layer (4, 3, 2) respectively. Since the previous illustration is confusing, we have improved it in the updated paper. At the mean time, as suggested by Reviewer #4, we add the layer index on Figure 1 for better clarity.
>
>
>
> [1] Neuhold G, Ollmann T, Rota Bulo S, et al. The mapillary vistas dataset for semantic understanding of street scenes[C]//Proceedings of the IEEE International Conference on Computer Vision. 2017: 4990-4999.
>
> [2] Godard C, Mac Aodha O, Firman M, et al. Digging into self-supervised monocular depth estimation[C]//Proceedings of the IEEE international conference on computer vision. 2019: 3828-3838.
>
> [3] Fu J, Liu J, Tian H, et al. Dual attention network for scene segmentation[C]//Proceedings of the IEEE Conference on Computer Vision and Pattern Recognition. 2019: 3146-3154.
>
> [4] Johnston A, Carneiro G. Self-supervised Monocular Trained Depth Estimation using Self-attention and Discrete Disparity Volume[C]//Proceedings of the IEEE/CVF Conference on Computer Vision and Pattern Recognition. 2020: 4756-4765.

---

### Official Review · AnonReviewer4 · 2020-10-28
**The paper presents a method for semantic-guided self-supervised depth estimation from monocular images. Attention and edge feature enhancement are the key contributions. Overall, good paper some minor clarifications required.**

**Rating:** 5
**Confidence:** 4

**Review:**

The paper presents a method for semantic-guided self-supervised depth estimation from monocular images. They propose semantic guidance to improve depth estimation performance. This is obtained by applying semantic guidance at multiple levels in the decoder via an attention layer and via feature enhancement in edges areas of the image. Experiments show the proposed method reaches state of the art performance for self-supervised monocular depth estimation in structure-from-motion setting.

Pros
- given semantic guidance, this method seems well designed and effective to enhance depth estimation performance with a relatively low computation overhead in order to provide semantic guidance
- interesting semantic guided edge point sampling strategy
- the paper shows an improved semantic guidance with respect to Guinzilini et al 2020b

Cons
- not clear how the decoder was designed, why on some layer there is attention and on other concatenation + convolution? Is it experimentally obtained, a design choice? Could the authors give some explanation about it?
- personally I find difficult to understand the choice of parameters \mu, c and N related to the semantic guided edge point sampling strategy, could the authors provide some detail about it?
- about the point feature extraction and enhancement, two unclear parts for me. First, "we select the 0 − 2 layers of features" could you explain what features/feature blocks you are referring to? Maybe a small explanation could be useful or adding in Fig. 1 some corresponding number to relate to. Second, "point feature enhancement module, which merges and enhances these point representations via a set of 1-D convolutions", the PFE is not clear, set of 1-D convolution means that for each channel of the feature map there is a convolution?
- Eq. 8, i \in [2,4] is related to the darker box around the decoder blocks in Fig. 1?
- the training set for Cityscape is the train split with fine annotations or also with coarse annotations?

Overall, the paper is well written, the results are sound and minor improvements could make it worth acceptance.

---

> ### Author Response · Authors · 2020-11-24
> **Response to Reviewer 4（ Part 1）**
>
> We thank the reviewer for the highly constructive and helpful suggestions to make the paper worth accepting. Here we address all the comments and update the corresponding information in the revised paper.
>
> **Q1:** Not clear how the decoder was designed, why on some layer there is attention and on other concatenation + convolution?
>
> **A1:** The design of attentions is based on the experimental analysis, in which we try to strike a balance between the effectiveness and computation efficiency of the proposed method. To this end, we fist train our model on KITTI with relative low input resolution (64 $\times$ 224), with varying attentions implemented on the decoding layers. The results are shown below.  Compared to model with a single attention (row 1), models with multi-level attentions show better performances. At the mean time, the performances of different multi-level attentions are close to each other (row 2, 3, 4). Consider the self-attention mechanism involves tensor multiplications which consume considerable GPU resources, we select the model with attentions on layer $[4,3,2]$ as the final choice. We added this part of illustration in Section A.3 of the Appendix.
>
> | Attentions on layers | AbsRel | SqRel | RMSE  | RMSElog | $\delta<1.25$ | $\delta<1.25^{2}$ | $\delta<1.25^{3}$ |
> | -------------------- | :----: | :---: | :---: | :-----: | :-----------: | :---------------: | :---------------: |
> | 4                    | 0.140  | 1.103 | 5.479 |  0.218  |     0.821     |       0.943       |       0.976       |
> | 4,3,2                | 0.136  | 1.070 | 5.441 |  0.214  |     0.830     |       0.945       |       0.977       |
> | 4,3,2,1              | 0.136  | 1.099 | 5.440 |  0.214  |     0.832     |       0.945       |       0.977       |
> | 4,3,2,1,0            | 0.137  | 1.065 | 5.391 |  0.214  |     0.828     |       0.945       |       0.977       |
>
>
>
> **Q2:** Personally I find difficult to understand the choice of parameters \mu, c and N related to the semantic guided edge point sampling strategy.
>
> **A2:**
>
> For parameter $N$ and $\mu$, inspired by previous works [1, 2], we set the ratio of edge-based points (including edge points $\mathcal{S_{E}}$ and disturb points $\mathcal{S_{D}}$) to random points $\mathcal{S_{R}}$ to be $4:1$ . In this way, the ratio parameter $\mu$ is set to $0.4$. At the mean time, we randomly select over $10000$ images from KITTI raw dataset to calculate the number of edge pixels, the mean value is $1233.01$. To make the number ($\mu N$) of the sampled edge points to be consistent with the real edge points number, we set $\mu N=1200$. Thus, the resulting number $N$ of all sampled points is set to $3000$. Under this setting, we find the sampled edge-based points $\mathcal{S_{E}}$ and $\mathcal{S_{D}}$ are able to cover most of the distinct edges, and the random points $\mathcal{S_{R}}$ are sampled uniformly on the whole image.
>
> For the choice of parameter $c$, we compute the ratio of the edge points which are close to the GT border within range $[-c,c]$. We use KITTI Semantics dataset for evaluation because it provides groundtruth KITTI semantic borders. As shown in the Table below, when $c$ is set to $3$, the corresponding ratio is 87.75%. This means the latent sampling areas of $\mathcal{S_{D}}$ has more than 87% overlappings with the real object borders. When $c$ is set to $5$, the ratio is 92.20%, which shows a relative small improvement (4.45%) toward that of $c=3$. At the mean time, since enlarging the range of the disturb points sampling will inevitably cut down the correlations between $\mathcal{S_{D}}$ and $\mathcal{S_{E}}$, we select $c=3$ as the final choice.
>
> We update the above illustrations to the paper in Section A.2 of the Appendix.
>
> | c           |   1   |   3   |   5   |
> | ----------- | :---: | :---: | :---: |
> | Ratio （%） | 55.30 | 87.75 | 92.20 |

---

> ### Author Response · Authors · 2020-11-24
> **Response to Reviewer 4（ Part 2）**
>
> **Q3:** In  the point feature extraction and enhancement, (1) could you explain what features/feature blocks you are referring to for " 0 − 2 layers" ? (2) The PFE is not clear, set of 1-D convolution means that for each channel of the feature map there is a convolution?
>
> **A3:**
>
> (1) Features on layer (2, 1, 0) refer to the 3 layers with largest feature resolutions of the encoding, semantic decoding and depth decoding branch, respectively (marked with darker boxes on Figure 1). We add the layer index on Figure 1 in the updated paper for better clarity.
>
> (2) The PFE module is used for enhancing the point-wise feature blocks via a set of convolutions. Since the 3-dimensional point feature block has the size of $[B,C_{J},N]$ ($B$ -- batch size, $C_{J}$ -- number of feature channels, $N$ -- number of sampled points), we do not leverage the conventional 2D-convolution that it is usually used for convolving 4-dimensional image feature blocks. Instead, we use the 1D-convolution which convolves the point feature block along the  third dimension -- the dimension of sampled N points. In this paper, we set the convolution kernel size to 1, thus the point-wise features will not be influenced by its spatial neighbors. We use the "Conv1d" function in Pytorch to implement our method.
>
>
>
> **Q4:** Eq. 8, i $\in$ [2,4] is related to the darker box around the decoder blocks in Fig. 1?
>
> **A4:** We didn't illustrate it clearly in the previous paper. The features of layer (4, 3, 2) are not included by the darker boxes, they actually refer to the first 3 decoding layers which has relative small feature resolutions.  After adding layer indexes in Figure 1 from the updated paper, we believe it is much easier to read now.
>
>
>
> **Q5:** The training set for Cityscapes is the train split with fine annotations or also with coarse annotations?
>
> **A5:** We do not train our network with Cityscapes dataset. The model is still trained on KITTI and we test it directly with Cityscapes test images. During testing, the binary semantic labels are required as input. So in Figure 6 and Figure 10 of the updated paper, we additionally compare the qualitative results guided by both pseudo and groundtruth semantic input. The results show comparable performances, which illustrate the advantage of our method that we do not need fine semantic labels as input for testing.
>
>
>
> [1] Kirillov A, Wu Y, He K, et al. Pointrend: Image segmentation as rendering[C]//Proceedings of the IEEE/CVF Conference on Computer Vision and Pattern Recognition. 2020: 9799-9808.
>
> [2]Xian K, Zhang J, Wang O, et al. Structure-Guided Ranking Loss for Single Image Depth Prediction[C]//Proceedings of the IEEE/CVF Conference on Computer Vision and Pattern Recognition. 2020: 611-620.

---

### Author Response · Authors · 2020-11-24
**General comments to reviewers**

We sincerely thank all reviewers for their valuable feedbacks. We are happy to see that the reviews found our paper "well designed and effective", "well written and the method clearly explained", "novel and interesting", "easy to read".

The main issues the reviewers raised are (1) the feasibility of semantic supervision, (2) more experimental results and (3) detailed explanations and clarifications of the method. We address these issues that (1) we experimentally demonstrate the feasibility of using off-the-shelf segmentation method to generate pseudo semantic labels with low costs, (2) we add extensive experiments on both main body of the paper and the Appendix to validate the effectiveness of our method and (3) we fix the typos and give detailed explanations of our method to make it easier to follow.

We have uploaded our revised paper, which we regard as a significantly improved version compared with the previous one. We add necessary illustrations and extensive experimental analyses trying to make this paper more technically sound. The updated parts are marked with blue for easy reading.

Due to the extensive experiments to conduct, our response comes a little late. We sincerely welcome the reviewers for discussions to further improve this paper.

---

### Decision · Program_Chairs · 2021-01-07
**Final Decision**

**Decision:**

Reject

**Comment:**

The authors address the problem of self-supervised monocular depth estimation via training with only monocular videos. They propose to use additional information extracted from semantic segmentation at training time to (i) provide additional “semantic context” supervision and (ii) to improve depth estimation at discontinuities through an edge guided point sampling based approach. Results are presented on the KITTI and Cityscapes datasets.

One of the main concerns is related to the utility of the semantic supervision given the relative cost required to obtain semantic training data in the first place. The authors state that "the pixel-wise local depth information can not be well represented by current depth network". However, Guizilini 2020a can generate detailed depth edges and they do NOT require any semantic information during training. The authors also state that "the required labeled semantic dataset only accounts for a very tiny proportion, which indicates a relatively lower cost." This is a bit misleading. The proposed method uses per pixel semantic ground truth from three datasets (Mapillary Vistas, Cityscapes, and KITTI). It takes a lot of effort to provide this ground truth compared to self-supervised methods which do not require any ground truth depth. It is encouraging that dataset specific semantic finetuning does not seem to have a large impact (Table 3), but this still requires access to a large enough initial semantic training set. Finally, the quantitative results are not much better than methods that don't require any semantics e.g. Guizilini 2020a, Johnston and Carneiro. Clearly, methods that do not require semantics are much more scalable, especially when adapting to new types of scenes.


Regarding the specific contributions of the paper, the SEEM module is the most novel component of the model. However, the addition of the SEEM module does not improve quantitative performance by much (see Table 2). In addition, the qualitative improvement it provides is also very subtle. This can be seen by comparing the last two rows of Fig 7 i.e. without and with. The authors need to make a stronger case, either quantitatively or qualitatively, as to why this is valuable.


Finally, but only a minor concern, the following relevant reference is missing: Jiao et al. Look Deeper into Depth: Monocular Depth Estimation with Semantic Booster and Attention-Driven Loss, ECCV 2018


In conclusion, there were mixed views from the reviewers - with some supportive of the paper (R2&3) others not as enthusiastic (R1&4). The authors should be commended for their detailed responses and changes already made based on reviewer comments and suggestions. Unfortunately, this did not change the mind of the reviewers. It is the opinion of this AC that there is still more work required to fully show the utility of the proposed approach, especially considering the non trivial effort that is required to obtain semantic supervision in novel domains.